# Learning Graph Foundation Models on Riemannian Graph-of-Graphs

**Haokun Liu** [* 1 2]  **Zezhong Ding** [* 3 2]  **Xike Xie** [1 2]

## Abstract

Graph foundation models (GFMs), pretrained on massive graph data, have transformed graph machine learning by supporting general-purpose reasoning across diverse graph tasks and domains. Existing GFMs pretrained with fixed-hop subgraph sampling impose a fixed receptive field, causing scale mismatch on diverse tasks, which often require heterogeneous and unknown structural contexts beyond a fixed sampling scale. We propose **R-GFM**, a Riemannian Graph-of-Graphs (GoG) based foundation model, that treats *structural scale* as a first-class citizen in modeling. R-GFM constructs a multi-scale GoG over-sampled subgraphs at different hop distances and learns geometry-adaptive representations from Riemannian manifolds. Theoretical analysis shows that R-GFM reduces structural domain generalization error compared to fixed-scale GFMs. Experiments on various datasets demonstrate that R-GFM achieves state-of-the-art performance, with up to a **49%** relative improvement on downstream tasks. Our code is available at https://github.com/USTC-DataDarknessLab/R-GFM.

## 1. Introduction

Graph foundation models (GFMs) (Wang et al., 2025b;a; Yu et al., 2025a; Liu et al., 2024; Xia & Huang, 2024), which are pretrained on massive graph datasets, have shown strong performance across diverse downstream applications, including molecular property prediction (Wu et al., 2017), quantum chemistry (Gilmer et al., 2017), web-scale recommendation (Ying et al., 2018; He et al., 2020), knowledge graph completion (Schlichtkrull et al., 2018), and spatio-

[*]Equal contribution [1]School of Biomedical Engineering, University of Science and Technology of China (USTC), Suzhou, Jiangsu, China [2]Data Darkness Lab, Suzhou Institute for Advanced Research, USTC, Suzhou, Jiangsu, China [3]School of Artificial Intelligence and Data Science, USTC, Hefei, Anhui, China. Correspondence to: Xike Xie <xkxie@ustc.edu.cn>.

*Proceedings of the 43rd International Conference on Machine Learning*, Seoul, South Korea. PMLR 306, 2026. Copyright 2026 by the author(s).

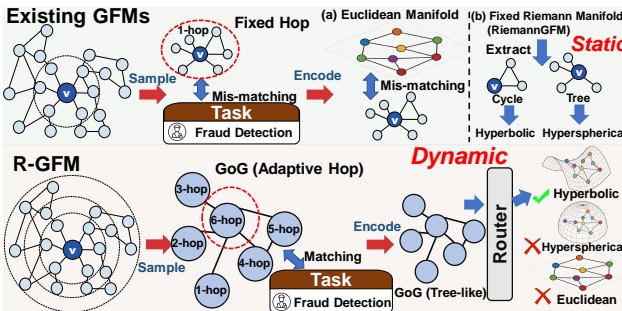

*Figure 1.* **Motivation of R-GFM.** Existing GFMs rely on fixed-hop sampling and static manifolds, causing representation mismatch. R-GFM constructs an adaptive-hop GoG and uses a router to dynamically choose the best-fit geometry for better matching.

temporal forecasting (Li et al., 2018).

Despite this progress, existing GFMs adopt a common design choice: they rely on *fixed-hop subgraph sampling* during pretraining, where each node or subgraph is encoded using a predetermined neighborhood range (e.g., 1-hop as shown in Figure 1). This design implicitly imposes a *fixed structural receptive field* on representation learning. As a result, the model is restricted to a static receptive field of structural context and fails to capture adaptive-hop structural information (i.e., diverse *structural scales*). However, in practice, different downstream tasks impose heterogeneous and unknown requirements on the hop range of subgraph sampling.

For instance, in node classification on highly homophilous citation networks (Kipf & Welling, 2016; Cavallo et al., 2022), low-hop information (e.g., within 2 hops) is often sufficient, as local neighbors tend to share similar labels. In contrast, for fraud detection in e-commerce (Liu et al., 2021), higher-hop information (e.g., beyond 4 hops) is required to uncover complex collusion patterns, where fraudsters hide their traces through long transaction chains.

More generally, fine-grained structural scales preserve feature distinctiveness (Han et al., 2022), but limit the scope of the receptive fields and are prone to the under-reaching problem (Alon & Yahav, 2021). In contrast, coarse-grained structural scales capture long-range dependencies (Lukovnikov & Fischer, 2021) and global structural information, but introduce irrelevant structural noise (Dong & Kluger, 2023) and

suffer from the over-smoothing problem (Keriven, 2022).

These observations reveal a fundamental *structural scale mismatch* in current GFMs: a fixed structural receptive field cannot simultaneously satisfy the heterogeneous structural requirements of diverse tasks, underscoring the necessity of designing a dynamic and adaptive, hop-aware representation learning paradigm.

In response, we propose R-GFM (as shown in Figure 1), a new GFM incorporating the adaptive-hop subgraphs of training nodes based on a dynamic Graph-of-Graphs (GoG) framework (Section 3.2). In R-GFM, subgraphs sampled at different hops are treated as nodes in a higher-order GoG, enabling the model to explicitly represent and reason over structural relationships across multiple scales. Different structural scales in the dynamically constructed GoGs often show different Riemannian geometric characteristics. Specifically, local-scale subgraphs tend to be dense and highly connected, while larger-scale subgraphs are typically sparse and hierarchical. Encoding such heterogeneous structures within a single Euclidean embedding space leads to geometric mismatch and representation distortion, which motivates a geometry-adaptive routing mechanism that assigns GoGs to appropriate Riemannian manifolds via a mixture-of-experts (MoE) framework.

However, routing GoGs to an appropriate Riemannian manifold poses non-trivial challenges. **First**, determining an appropriate number of Riemannian experts is non-trivial. Too few experts lead to underfitting, while too many experts result in overfitting and training instability. **Second**, existing graph MoE models (Xia & Huang, 2024) typically select a fixed top-$m$ set of experts for aggregation. An overly small $m$ leads to insufficient representation capacity, while an excessively large $m$ exacerbates generalization (Zhao et al., 2024b), thereby hindering model convergence. In response, we propose a dynamic MoE-based Riemannian manifold routing strategy (Section 3.3) that maintains a dynamic candidate set of Riemannian experts and activates the experts for each input GoG, capturing the Riemannian geometric information in GoGs and improving generalization.

These designs are supported by both theoretical analysis and empirical evaluation. Theoretically, we prove that our adaptive-hop GoG construction reduces subgraph embedding error compared to fixed-hop. Moreover, we prove that our routing strategy yields a tighter excess-risk bound and a tighter cross-domain generalization error bound than using a fixed expert set. Further experiments demonstrate that R-GFM shows strong domain generalization on diverse real-world graphs and downstream tasks (Section 4).

Our main contributions are summarized as follows:

- We propose R-GFM, a novel GFM based on the Graph-of-Graphs mechanism, enabling scale-adaptive domain generalization across diverse graph domains.

- We introduce a memory-aware GoG construction strategy that effectively captures structural information across varying hops with theoretical guarantees.

- We introduce an MoE-based Riemannian routing strategy that effectively captures Riemannian geometric information in GoGs with theoretical guarantees.

- Experiments on **18** real-world graphs and various downstream tasks demonstrate consistent superiority over baselines, achieving up to a **49%** improvement.

## 2. Preliminaries

### 2.1. Attributed Graph

We denote an attributed graph as $G = (V, E, \mathbf{X}_V)$ with node set $V$, edge set $E$, and node features $\mathbf{X}_V$.

$\text{dist}(u, v)$ denotes the shortest-path distance between nodes $u \in V$ and $v \in V$. $\mathcal{N}_k(u) = \{v \in V : \text{dist}(u, v) \leq k\}$ denotes the $k$-hop neighbors of node $u$.

### 2.2. GFM Design Problem (Wang et al., 2025a)

Given a set of graphs $\{G_i\}$ (including training and testing graphs) from domains $\mathcal{D} = \{D_i\}$, our goal is to design a GFM that is pretrained only on the training graph domains $D_{\text{Train}} \subset \mathcal{D}$. The objective is to maximize the downstream task performance (e.g., node classification) on the test graphs from the unseen domain $D_{\text{Test}} \cap D_{\text{Train}} = \emptyset$.

### 2.3. Graph-of-Graphs (GoG) (Zhen et al., 2023)

Given a graph set $\{G_i\}$, the GoG of $\{G_i\}$ is denoted as $\mathcal{G} = (\mathcal{V}, \mathcal{E}, \mathbf{X}_G)$, where $\mathcal{V} = \{G_i\}$, $\mathcal{E}^1 \subseteq \mathcal{V} \times \mathcal{V}$, and $\mathbf{X}_G$ is the graph features of $\{G_i\}$.

### 2.4. Riemannian Manifold (Sun et al., 2025)

A Riemannian manifold is a geometric space, which is a smooth manifold endowed with a Riemannian metric. The curvature $\kappa$ is the geometric quantity measuring the extent to which a surface deviates from being flat. Following the setting of (Sun et al., 2025), we use the constant-curvature Riemannian manifolds, where curvature $\kappa$ is equal everywhere, including three types of manifolds: (1) Hyperbolic manifold with $\kappa < 0$, (2) Euclidean manifold with $\kappa = 0$, and (3) Hyperspherical manifold with $\kappa > 0$.

---

[1] In our design, $\mathcal{E}$ is a subset of $\mathcal{V} \times \mathcal{V}$, which is constructed by graph similarity-based sampling strategy, as shown in Equation 3.

# 3. The Proposed GFM: R-GFM

In this section, we present R-GFM, a GFM that combines adaptive-hop GoG construction with an MoE-based Riemannian routing to produce transferable node representations.

## 3.1. Overview of R-GFM

In Figure 2, the R-GFM framework consists of four stages. **Stage A (Steps ①-②)** determines the candidate Riemannian expert set and samples subgraphs of each training node with hop distances ranging from 1 to $K$. **Stage B (Step ③)** encodes the sampled subgraphs to obtain their embeddings. **Stage C (Steps ④-⑤)** constructs a GoG based on subgraph embeddings and applies MoE-based Riemannian routing to encode the GoG by using the active Riemannian experts, generating GoG embeddings. **Stage D (Step ⑥)** aggregates the GoG embeddings into a single fused embedding, which is used for downstream tasks. In the sequel, we present two core strategies in Steps ④-⑤, including adaptive-hop GoG Construction and MoE-based Riemannian routing in Sections 3.2 and 3.3, respectively.

## 3.2. Adaptive-Hop GoG Construction

We now describe how to construct a GoG for each training node to capture different structural scales. The construction proceeds as follows: (1) determining the GoG node set via multi-hop subgraph sampling, and (2) establishing GoG edges based on subgraph similarity.

**Memory-aware GoG Node Determination.** To capture both local and global information, we sample subgraphs ($\{G_v^{(i)}\}_{i=1}^K$) centered at node $v$ from hop distances 1 to $K$, where $K$ is a dynamic parameter that controls the receptive field size and determines the number of GoG nodes. Larger $K$ enables richer structural modeling but incurs higher GPU memory cost. Therefore, our objective is to maximize $K$ under an explicit GPU memory budget $\mathcal{B}_{\mathrm{GPU}}$.

To address the problem, we design an online greedy strategy. We progressively enlarge $K$ and test whether the resulting configuration fits within $\mathcal{B}_{\mathrm{GPU}}$. Once the feasibility test fails (e.g., out-of-memory), we roll back and return the largest feasible $K$.

**Subgraph Similarity-based GoG Edge Construction.** After determining the GoG node set, we construct GoG edges based on subgraph similarity. We define subgraph similarity as follows.

**Definition 3.1** (**Subgraph Similarity**). The *similarity* between two subgraphs is defined as the cosine similarity of their embeddings. The *subgraph similarity matrix* (**S**) includes all pairwise subgraph similarities as:

$$\mathbf{S} = \mathbf{X}_{\mathrm{sub}}\big(\mathbf{X}_{\mathrm{sub}}\big)^\top \in \mathbb{R}^{K \times K}, \tag{1}$$

where $\mathbf{X}_{\mathrm{sub}}$ is the subgraph embedding matrix.

Rather than densely connecting all subgraphs, we construct a sparse GoG by sampling edges according to subgraph similarity. Specifically, we normalize the similarity scores into a sampling distribution:

$$\mathrm{Prob}(i,j) = \frac{e^{\mathbf{S}[i,j]}}{\sum_u \sum_v e^{\mathbf{S}[u,v]}}, \tag{2}$$

where $\mathrm{Prob}(i,j)$ is the probability of sampling edge $(i,j)$, e is Euler's number.

We then sample $\mathcal{B}_{\mathrm{edge}}$ edges without replacement[2], as shown in Equation 3.

$$\mathcal{E} \sim \mathrm{Sample}\big(\mathrm{Prob}; \mathcal{B}_{\mathrm{edge}}\big), \text{subject to } |\mathcal{E}| = \mathcal{B}_{\mathrm{edge}} \tag{3}$$

where $\mathcal{E}$ is the GoG edge set.

Finally, we symmetrize the sampled edges to enable bidirectional message passing, which empirically improves structural pattern identification and representation quality (Egressy et al., 2024).

**Theoretical Analysis.** We provide theoretical justification for our GoG construction strategy from two perspectives: (1) why sampling subgraphs from varied hops is preferable to using a fixed hop (Theorem 3.2), and (2) why constructing GoG edges improves subgraph representation quality. (Theorem 3.3).

To begin, we first introduce the concept of *embedding noise* and the *sampled subgraph embedding* to analyze the impact of embedding noise on subgraph representations (Li et al., 2024). Typically, the subgraph embedding ($\mathbf{x}$) follows a Gaussian distribution ($\mathcal{N}$) with noise level $\sigma$, where smaller $\sigma$ corresponds to higher representation quality (Li et al., 2024)), as shown in Equation 4.

$$\mathbf{x} \sim \mathcal{N}(\boldsymbol{\mu}, \boldsymbol{\sigma}^2), \tag{4}$$

where $\boldsymbol{\mu}$ is noise-free subgraph embedding.

Let $\boldsymbol{\sigma}_{\mathrm{F}}$ denote the embedding noise induced by sampling from a fixed hop, and let $\boldsymbol{\sigma}_{\mathrm{V}}$ denote the embedding noise induced by sampling across multiple hops under our strategy. Theorem 3.2 follows from these definitions.

**Theorem 3.2.** *The embedding noise of sampling from various hops is lower than that of sampling from a fixed hop:* $\|\boldsymbol{\sigma}_{\mathrm{V}}\|_2 \leq \|\boldsymbol{\sigma}_{\mathrm{F}}\|_2$, *where* $\|\cdot\|$ *denotes the* $\ell_2$ *norm.*

The proof of Theorem 3.2 is in Appendix G.

Theorem 3.2 implies that sampling subgraphs across multiple hops yields lower embedding noise than fixed-hop sampling, thereby validating the effectiveness of our sampling strategy.

---

[2]We set $\mathcal{B}_{\mathrm{edge}} = 0.6 \times \frac{K(K-1)}{2}$, as discussed in Section 4.6.

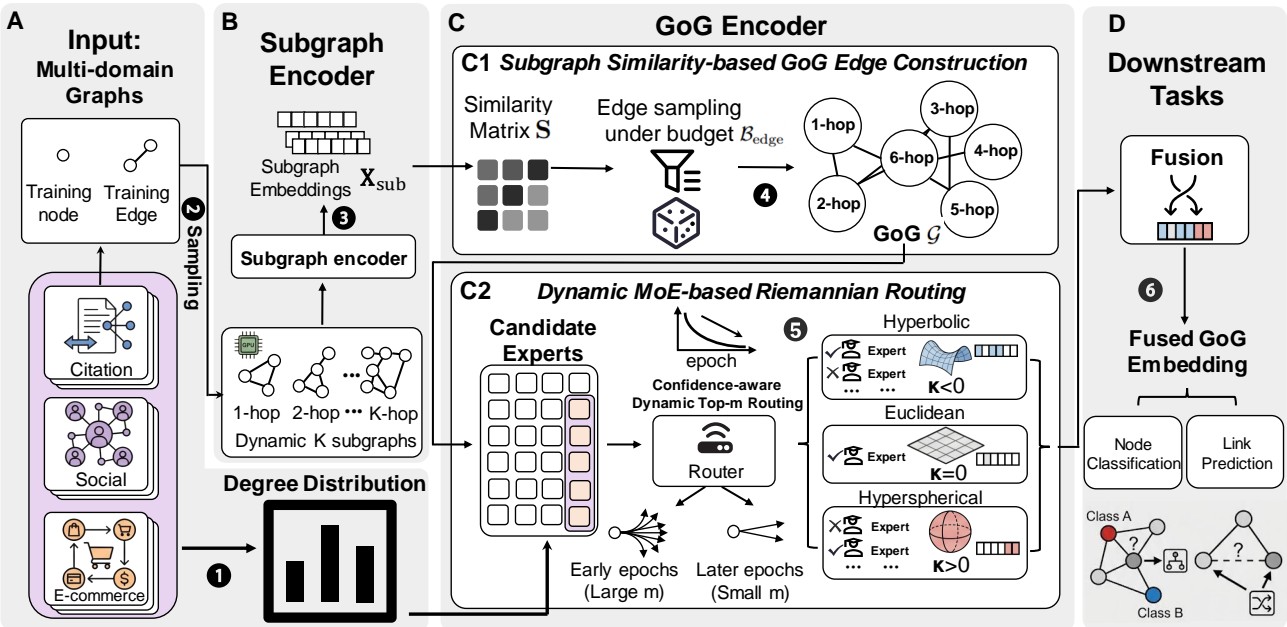

*Figure 2.* **Overview of R-GFM.** ① Calculate the coefficient of variation (CV) to quantify the node degree distribution, which is used to determine the candidate Riemannian expert set. ② Sample adaptive-hop subgraphs for each training node $v$ (or an edge $(u, v)$ in link prediction task (Zhang & Chen, 2018)) with a adaptive-hop strategy (Section 3.2). ③ Encode sampled subgraphs $\{G_v^{(i)}\}_{i=1}^K$ to obtain subgraph embeddings ($\mathbf{X}_{\mathrm{sub}}$), and pretrain this subgraph encoder via contrastive learning using NT-Xent loss (Chen et al., 2020). ④ Construct a GoG $\mathcal{G}$ by sampling edges from the subgraph similarity matrix $\mathbf{S}$ under the edge number budget $\mathcal{B}_{\mathrm{edge}}$ (Section 3.2). ⑤ Encode $\mathcal{G}$ with the dynamic MoE-based Riemannian routing encoder and perform dynamic top-$m$ expert routing (Section 3.3). ⑥ generate the fused GoG embedding, which is used for downstream tasks, such as node classification or link prediction.

We next analyze the effectiveness of our GoG edge construction strategy by comparing it with alternative GoG edge designs under an identical set of GoG nodes. Specifically, we show that our similarity-based sparse construction achieves strictly lower expected embedding error than two canonical constructions: a GoG without edges and a fully connected GoG. These constructions correspond to the extreme cases of no cross-scale interaction and indiscriminate interaction, respectively, thereby demonstrating the advantage of selective structural connectivity as shown in Theorem 3.3.

**Theorem 3.3** (**Effectiveness of GoG Edge Construction**). *Consider GoG constructions that share the same set of GoG nodes. Let $e_{\mathrm{none}}$, $e_{\mathrm{full}}$, and $e_{\mathrm{ours}}$ denote the squared embedding error obtained by (i) a GoG without edges, (ii) a full connected GoG, and (iii) our similarity score-based GoG, respectively. Then, $e_{\mathrm{ours}} < e_{\mathrm{none}}$ and $e_{\mathrm{ours}} < e_{\mathrm{full}}$.*

The proof of Theorem 3.3 is in Appendix H.

### 3.3. Dynamic MoE-based Riemannian Routing

Given a constructed GoG $\mathcal{G} = (\mathcal{V}, \mathcal{E}, \mathbf{X}_{\mathrm{G}})$, we learn its representation using a dynamic MoE-based Riemannian routing strategy. Our routing is dynamic in two aspects: (1) the candidate curvature set $\mathcal{K}$ is dynamically selected to determine the candidate Riemmanian expert set $M$, based

on the node degree distributions of given datasets; and (2) the number of activated experts is adjusted dynamically via confidence-aware Top-$m$ routing during training.

**Dynamical Candidate Expert Set Determination.** Motivated by prior work showing that heterogeneous graph structures benefit from diverse geometric inductive biases (Gu et al., 2019), we adopt a structure-aware strategy based on node degree distributions (Hu et al., 2017). We quantify structural heterogeneity using the coefficient of variation (CV):

$$\mathrm{CV}(\mathcal{D}_i) = \frac{\mathrm{std}(\deg(\mathcal{D}_i))}{\mathrm{mean}(\deg(\mathcal{D}_i))}, \qquad (5)$$

where $\deg(\mathcal{D}_i)$ is the node degrees of graph dataset $\mathcal{D}_i$.

Then, we define the score $\mathcal{S}_i$ to quantify the structural heterogeneity of datasets $\{\mathcal{D}_1, \mathcal{D}_2, \cdots, \mathcal{D}_i\}$ as:

$$\mathcal{S}_i = \mathrm{normalize}(\mu_i + \sigma_i), \qquad (6)$$

where $\mu_i$ is the mean of $\{\mathrm{CV}(\mathcal{D}_1), \cdots, \mathrm{CV}(\mathcal{D}_i)\}$ and $\sigma_i$ is the standard variance of that. Here, $\mu_i$ captures the average structural heterogeneity across datasets, while $\sigma_i$ measures its variability. We apply normalization to ensure scale consistency.

Then, based on the score, we design a candidate expert set

determination strategy as shown below. Initially, we start with a single expert and predefine a Riemannian curvature list, where each $\kappa_i$ corresponds to the curvature of the $i$-th candidate expert.

As datasets $\{\mathcal{D}_1, \mathcal{D}_2, \cdots\}$ are introduced sequentially, we update the heterogeneity score $\mathcal{S}_i$ upon receiving each new dataset according to Equation 6. Before training on $\{\mathcal{D}_1, \mathcal{D}_2, \cdots, \mathcal{D}_i\}$, we compute $\mathcal{S}_i$ and determine the candidate set size as $\lceil \mathcal{S}_i \cdot \zeta \rceil$. After that, we generate the set of curvature values by starting at $0$ and alternating between decreasing and increasing integer values (e.g., $0, -1, +1, -2, +2, \ldots$) until the set reaches size $\lceil \mathcal{S}_i \cdot \zeta \rceil$. Then, we use this curvature set to determine the candidate Riemannian experts. Each curvature defines a specific Riemannian manifold, which is associated with one expert.

**Confidence-aware Dynamic Top-$m$ Routing.** Standard MoE models select a fixed Top-$m$ experts for routing (Shazeer et al., 2017). In contrast, we dynamically adjust $m$ based on routing confidence.

To begin, we define the routing score of a $\mathcal{G}$ in Equation 7.

$$\boldsymbol{\alpha}_{\mathcal{G}} = \mathrm{softmax}(g(\mathcal{G})/\tau), \tag{7}$$

where $\tau$ is a temperature. We use GCN (Kipf & Welling, 2016) as the GNN encoder $g(\cdot)$, following the setting of (Guo et al., 2025). $\boldsymbol{\alpha}_{\mathcal{G}} \in \mathbb{R}^{\psi \times 1}$, where $\psi$ is the number of experts.

Then we define a global confidence score in Equation 8.

$$\mathrm{conf} = \frac{1}{\psi} \sum_{1 \leq i \leq \psi} \max \alpha_{\mathcal{G}}^{(i)}, \tag{8}$$

where $\alpha_{\mathcal{G}}^{(i)}$ is the routing score of expert $i$.

Since the router confidence typically grows during training, the router becomes more confident (i.e., producing more peaked distributions). This implies that fewer experts are sufficient to capture GoG structure information.

Motivated by this (Huang et al., 2024a), we design the workflow of updating $m$ as follows. Initially, $m$ is set to an initial value $m_0$. Then, we dynamically adjust the active expert number $m$ at the epoch level by decreasing it stepwise: $m \leftarrow \max(1, m - \mathrm{conf})$.

**Theoretical Analysis.** Here, we theoretically analyze our strategies from these two perspectives: (1) why a dynamic candidate expert set is needed for different input datasets (Theorem 3.4) and (2) why our method theoretically outperforms MDGFM (Wang et al., 2025a), which is the state-of-the-art GFM with theoretical analysis (Theorem 3.5).

**Theorem 3.4** (**Excess-risk Upper Bound Analysis**). *Considering $N$ training datasets $\{\mathcal{D}_1, \mathcal{D}_2, \cdots, \mathcal{D}_N\}$, let the*

*excess risk (Koltchinskii, 2010) upper bound of selected expert number with $j$ experts as $\mathcal{R}(j)$:*

$$\mathcal{R}(j) = \frac{A\,\mathcal{S}_N}{j} + B\sqrt{\frac{j}{n_N}}, \tag{9}$$

*where $A, B > 0$ are constants, $n_N$ is the number of constructed GoGs from $\mathcal{D}_1$ to $\mathcal{D}_N$, and $S_N$ is the score defined in Equation 6.*

*Let $\psi_F$ be the fixed candidate expert number, and let $\psi_D$ be our dynamically selected expert number. Then:*

$$\mathcal{R}(\psi_D) \leq \mathcal{R}(\psi_F). \tag{10}$$

*Equality holds if and only if the fixed number of experts is identical to the number determined by our dynamic strategy.*

Typically, existing methods (Guo et al., 2025) employ a manually specified number of experts, which requires extensive trial-and-error tuning and fails to adapt to various graph datasets (i.e., the equality in Equation 10 does not hold in general). The proof of Theorem 3.4 is in Appendix I.

**Theorem 3.5** (**Domain Generalization Error Bound Analysis**). *Let $\Phi_{\mathrm{M}}$ denote the encoder class induced by MDGFM (Wang et al., 2025a), and let $\Phi_{\mathrm{R}}$ denote the encoder class induced by R-GFM. Let the best achievable target-domain surrogate upper bounds of MDGFM and R-GFM be $\epsilon_{\mathrm{MDGFM}}, \epsilon_{\mathrm{R\text{-}GFM}}$. Under the assumptions detailed in Appendix J, we have $\epsilon_{\mathrm{R\text{-}GFM}} < \epsilon_{\mathrm{MDGFM}}$.*

The proof of Theorem 3.5 is in Appendix J.

## 4. Experiments

In this section, we conduct extensive experiments to answer the following research questions:
(**RQ1**): Can R-GFM outperform existing models on downstream tasks under the same training settings?
(**RQ2**): How robust is R-GFM to graph perturbations compared with existing models?
(**RQ3**): How do the key components of R-GFM contribute to its performance?
(**RQ4**): How does R-GFM scale?
(**RQ5**): How should we choose the key hyperparameters of R-GFM in practice, and how do they affect performance?

### 4.1. Experimental Setup

**Datasets** In our main setting, we use 10 benchmark graphs from 4 domains, which are widely used in prior GFM studies (Zhao et al., 2024a; Wang et al., 2025a; Yu et al., 2025b), as shown in Table 7 of Appendix C.1. We adopt a leave-one-dataset-out transfer setting: for each target dataset, we pretrain on the other datasets and test on the held-out target. To further validate R-GFM under a unified semantic feature space and at larger scale, we additionally use

4 large training datasets including ArXiv_2023 (He et al., 2023), ogbn-Arxiv (Wang et al., 2020b), Reddit (Huang et al., 2024b), and PubMed (Yang et al., 2016), and 4 test datasets including Cora (Sen et al., 2008), Ele-Computers and Books-History (Yan et al., 2023), and Instagram (Huang et al., 2024b). More information about the datasets is provided in Appendix C.2.

**Baselines.** As summarized in Table 1, we group all baselines into four categories based on their training and adaptation paradigms: (1) **Task-Supervised GNNs**, (2) **Self-Supervised Pretraining with Fine-Tuning**, (3) **Prompt-based Adaptation**, and (4) **Graph Foundation Models**. More information is provided in Appendix E.1.

**Evaluation Metrics** We evaluate R-GFM on two task types: node classification and link prediction. Following the metric setting in (Sun et al., 2025), we use accuracy in node classification and AUC-ROC in link prediction.

### 4.2. Downstream Task Performance (RQ1)

We address (**RQ1**) by evaluating on diverse downstream tasks, including node classification and link prediction.

**Node Classification.** We report the 1-shot results in Table 1. The 3-shot and 5-shot results are in Tables 2 and 3. Under different shot settings, R-GFM consistently achieves the best performance on all datasets, demonstrating strong cross-domain generalization. Notably, R-GFM yields large gains on citation networks and WebKB-style graphs (e.g., 18.9% improvement on Citeseer).

We additionally evaluate R-GFM in a large-scale setting using unified language features described in Section 4.1. The results are summarized in Table 4. Overall, R-GFM remains the best performance under this setting.

**Link Prediction** We evaluate on link prediction via full-parameter fine-tuning in Table 5, following the task setting in (Sun et al., 2025).

More results are in Tables 9 of Appendix D.1. Overall, R-GFM achieves consistently strong AUC-ROC on these representative datasets.

### 4.3. Robustness to Perturbations (RQ2)

To answer **RQ2**, we evaluate R-GFM under two graph perturbations: edge drop and node masking. Based on Appendix D.1, we apply perturbations only at the evaluation phase and report how performance changes as the perturbation magnitude increases.

**Edge Drop.** We randomly remove edges with drop rate $p \in \{0, 0.1, 0.2, 0.3, 0.4, 0.5\}$. In Figure 3a, R-GFM con-

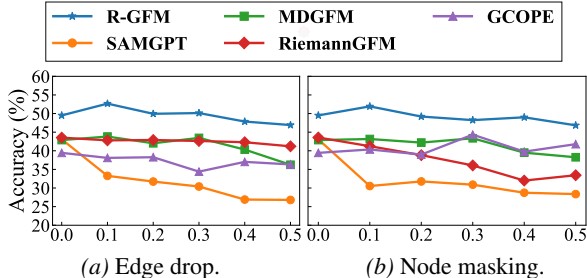

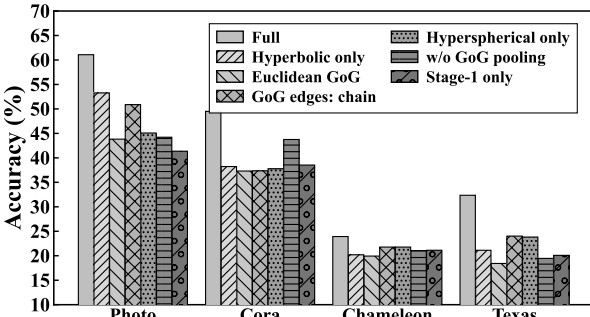

*Figure 3.* Robustness to structural and attribute perturbations.

*Figure 4.* Ablation results on four representative datasets (Photo, Cora, Chameleon, Texas) under 1-shot node classification.

sistently achieves the best accuracy across all perturbation levels and exhibits the slowest degradation rate as $p$ increases. For example, at $p = 0.5$, R-GFM retains 46.91% (a drop of 5.76% points from $p = 0$), while the strongest baseline drops to around 41.2%.

**Node Masking.** We randomly mask node attributes with masking ratio $r \in \{0, 0.1, 0.2, 0.3, 0.4, 0.5\}$ by setting the corresponding feature vectors to zero. Figure 3b shows a similar trend: R-GFM maintains stable performance under increasing attribute missingness. At $r = 0.5$, R-GFM retains 46.84% accuracy, whereas the strongest baseline drops to 41.78%, resulting in a 5.06% advantage.

### 4.4. Ablation Study (RQ3)

To answer **RQ3**, we ablate the two key design choices of R-GFM: (i) constructing the GoG, and (ii) employing Riemannian encoders on the GoG. We report results on four datasets, each from a different domain, to examine the effect across diverse domains. As shown in Figure 4, removing or weakening the GoG module (Stage-1 only, w/o GoG pooling, or using a GoG topology with chain edges) consistently hurts performance, verifying the effectiveness of GoG construction. Moreover, replacing the Riemannian encoder on GoG with an Euclidean one (Euclidean GoG) leads to a clear drop across datasets, and using only one manifold (Hyperbolic only or Hyperspherical only) is inferior to the full model, indicating the benefit of Riemannian MoE.

*Table 1.* Accuracy Evaluation on 1-shot Node Classification.

| Methods | Wisconsin | Cornell | Citeseer | Cora | Pubmed | Computers | Photos | Texas |
|---|---|---|---|---|---|---|---|---|
| **Task-Supervised GNNs** | | | | | | | | |
| GCN | 17.46 ± 3.35 | 19.53 ± 5.77 | 26.89 ± 3.01 | 31.98 ± 4.11 | 44.29 ± 2.22 | 39.43 ± 5.68 | 50.39 ± 1.19 | 18.48 ± 4.08 |
| GAT | 16.86 ± 3.16 | 16.51 ± 8.41 | 25.27 ± 2.10 | 26.81 ± 4.42 | 45.11 ± 5.52 | 38.05 ± 4.47 | 56.51 ± 2.35 | 18.36 ± 4.63 |
| H2GCN | 20.34 ± 5.77 | 29.30 ± 10.40 | 24.47 ± 1.35 | 23.61 ± 3.92 | 40.97 ± 1.26 | 19.29 ± 6.87 | 42.95 ± 4.32 | 29.94 ± 16.38 |
| FAGCN | 26.19 ± 7.93 | 28.72 ± 8.12 | 20.56 ± 1.02 | 20.75 ± 3.85 | 41.65 ± 1.46 | 27.02 ± 2.97 | 47.91 ± 2.17 | 31.35 ± 14.17 |
| **Self-Supervised Pretraining with Fine-tuning** | | | | | | | | |
| DGI | 28.24 ± 5.30 | 29.73 ± 10.11 | 34.52 ± 8.81 | 39.88 ± 5.53 | 47.10 ± 4.95 | 34.59 ± 5.94 | 50.66 ± 6.93 | 29.73 ± 6.34 |
| GraphCL | 32.16 ± 8.83 | 31.35 ± 12.47 | 27.56 ± 5.01 | 35.26 ± 8.62 | 43.66 ± 3.12 | 34.40 ± 6.93 | 48.86 ± 5.53 | 31.35 ± 22.66 |
| GraphACL | 29.41 ± 11.09 | 25.41 ± 9.86 | 28.34 ± 4.19 | 36.96 ± 8.71 | 38.20 ± 10.11 | 44.42 ± 7.06 | 56.60 ± 8.90 | 24.86 ± 9.44 |
| MixHop | 29.82 ± 9.32 | 17.14 ± 5.54 | 21.35 ± 3.67 | 21.46 ± 5.95 | 43.57 ± 4.05 | 29.77 ± 15.57 | 32.77 ± 9.26 | 26.09 ± 15.93 |
| **Prompt-based Adaptation** | | | | | | | | |
| GPPT | 23.53 ± 15.93 | 25.41 ± 5.27 | 33.24 ± 8.73 | 40.60 ± 6.00 | 41.64 ± 9.29 | 44.57 ± 13.38 | 53.94 ± 11.93 | 18.38 ± 12.00 |
| GPF | 23.53 ± 9.36 | 36.22 ± 9.46 | 27.16 ± 4.10 | 34.74 ± 3.48 | 26.62 ± 10.18 | 42.15 ± 8.75 | 59.13 ± 6.05 | 28.65 ± 17.31 |
| GraphPrompt | 20.39 ± 7.92 | 28.65 ± 5.92 | 19.80 ± 2.79 | 33.68 ± 9.83 | 42.94 ± 1.98 | 33.34 ± 6.09 | 42.60 ± 6.75 | 24.32 ± 9.17 |
| GraphPrompt+ | 16.86 ± 5.63 | 29.19 ± 5.24 | 28.28 ± 3.55 | 29.76 ± 6.06 | 38.74 ± 1.70 | 43.01 ± 6.72 | 58.23 ± 4.61 | 19.46 ± 14.84 |
| **Graph Foundation Models** | | | | | | | | |
| GCOPE | 28.69 ± 7.00 | 25.00 ± 6.58 | 38.62 ± 2.58 | 39.05 ± 2.53 | 44.52 ± 2.51 | 28.50 ± 4.91 | 59.17 ± 2.45 | 19.50 ± 12.18 |
| SAMGPT | 21.57 ± 12.09 | 36.22 ± 5.30 | 28.78 ± 8.99 | 43.08 ± 10.40 | 37.95 ± 9.58 | 45.80 ± 4.54 | 49.80 ± 11.89 | 31.89 ± 11.26 |
| MDGFM | 17.65 ± 7.34 | 26.49 ± 10.73 | 23.32 ± 15.32 | 42.88 ± 9.12 | 38.42 ± 7.92 | 42.68 ± 2.53 | 57.78 ± 9.07 | 31.54 ± 11.78 |
| RiemannGFM | 31.20 ± 9.35 | 31.35 ± 7.76 | 31.86 ± 6.45 | 43.48 ± 5.28 | 44.10 ± 5.05 | 46.20 ± 8.79 | 57.20 ± 16.84 | 29.12 ± 7.44 |
| GPM | 29.19 ± 4.69 | 27.83 ± 11.00 | 29.88 ± 6.36 | 38.96 ± 8.18 | 35.80 ± 8.20 | 48.58 ± 10.66 | 37.85 ± 8.17 | 27.70 ± 12.09 |
| G2PM | 34.70 ± 8.73 | 32.96 ± 5.44 | 31.04 ± 2.90 | 40.04 ± 3.13 | 49.71 ± 8.66 | 51.92 ± 10.10 | 59.90 ± 5.02 | 30.37 ± 9.36 |
| R-GFM | **35.41 ± 7.29** | **36.71 ± 9.92** | **57.54 ± 9.49** | **49.50 ± 3.97** | **49.80 ± 5.38** | **52.30 ± 3.33** | **61.08 ± 5.26** | **32.36 ± 12.10** |

Results are reported in percent. The best method is bolded, and the runner-up is underlined.

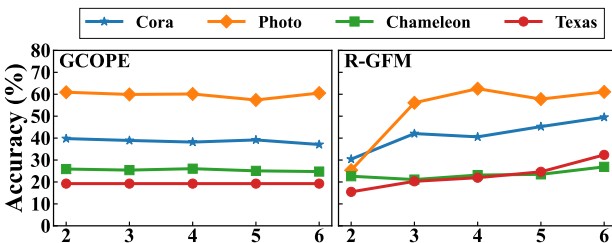

*Figure 5.* Performance scaling with hop number $K$: a naive $K$-hop extension of GCOPE vs. R-GFM under matched settings.

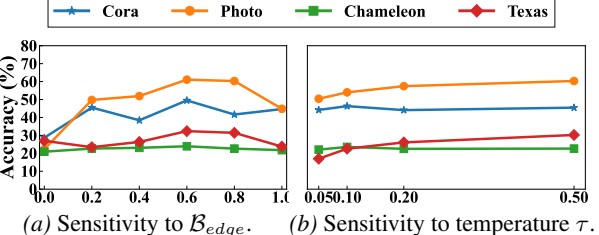

*(a)* Sensitivity to $\mathcal{B}_{edge}$.    *(b)* Sensitivity to temperature $\tau$.

*Figure 6.* Hyperparameter sensitivity on four datasets. Left: varying $\mathcal{B}_{edge}$. Right: varying temperature $\tau$.

### 4.5. Scalability (RQ4)

To answer **RQ4**, we test scalability from two perspectives: (1) scaling with structural scales (i.e., increasing $K$-hop), and (2) scaling with graph size (i.e., larger graphs), reporting both accuracy and resource costs.

**Scaling with Hop Number ($K$)** We study how performance scales with the hop number $K$ on cross-domain 1-shot node classification by extending GCOPE to sample subgraphs of varying hops. In Figure 5, simply increasing $K$ yields only marginal gains for the naive GCOPE extension, suggesting that enlarging the receptive field alone is insufficient. In contrast, R-GFM benefits more consistently from larger $K$, indicating that our GoG-based multi-scale construction can better leverage additional hop information.

**Scaling with Graph Size.** Next, we evaluate R-GFM on

larger graphs under practical budgets with results shown in Table 4. Under this setting, SAMGPT and MDGFM hit out-of-memory (OOM) due to RAM exhaustion, as expected given their full-graph training pipeline. In contrast, R-GFM leverages subgraph sampling, which enables it to scale well to larger graphs under the same memory budget. It also delivers the best transfer performance on the downstream datasets, demonstrating strong scalability in both efficiency and effectiveness.

### 4.6. Hyperparameter Study(RQ5)

To answer **RQ5**, we conduct a hyperparameter study and report the 1-shot performance across datasets.

**GoG Edge Budget $\mathcal{B}_{edge}$.** Since the node number $K$ in GoG is dynamic, GoG does not have a fixed number of

*Table 2.* Accuracy evaluation on **3-shot** node classification under the protocol in Section 4.1.

| Methods | Wisconsin | Cornell | Citeseer | Cora | Pubmed | Computers | Photos | Texas |
|---|---|---|---|---|---|---|---|---|
| **Task-Supervised GNNs** | | | | | | | | |
| GCN | $15.51 \pm 5.61$ | $20.00 \pm 2.04$ | $33.61 \pm 4.98$ | $40.12 \pm 6.52$ | $40.97 \pm 2.16$ | $31.54 \pm 5.47$ | $59.05 \pm 4.60$ | $21.95 \pm 11.22$ |
| GAT | $18.94 \pm 1.99$ | $15.46 \pm 5.51$ | $22.07 \pm 8.64$ | $29.20 \pm 3.13$ | $34.70 \pm 7.32$ | $38.92 \pm 9.21$ | $47.20 \pm 4.62$ | $28.05 \pm 12.06$ |
| H2GCN | $21.32 \pm 13.55$ | $31.17 \pm 11.47$ | $23.29 \pm 2.97$ | $24.94 \pm 6.05$ | $42.95 \pm 3.02$ | $13.77 \pm 4.30$ | $35.97 \pm 4.50$ | $16.95 \pm 1.36$ |
| FAGCN | $31.32 \pm 5.57$ | $31.90 \pm 11.89$ | $20.15 \pm 0.65$ | $19.75 \pm 6.68$ | $40.09 \pm 1.09$ | $25.51 \pm 5.75$ | $42.35 \pm 2.53$ | $26.95 \pm 8.06$ |
| **Self-Supervised Pretraining with Fine-tuning** | | | | | | | | |
| DGI | $36.08 \pm 3.82$ | $30.27 \pm 14.48$ | $45.26 \pm 2.85$ | $58.26 \pm 2.61$ | $52.94 \pm 7.65$ | $45.15 \pm 6.63$ | $68.87 \pm 9.46$ | $40.00 \pm 5.54$ |
| GraphCL | $36.08 \pm 10.61$ | $44.32 \pm 13.46$ | $32.02 \pm 3.48$ | $45.30 \pm 6.52$ | $44.20 \pm 1.53$ | $49.62 \pm 7.42$ | $62.69 \pm 0.12$ | $41.62 \pm 10.22$ |
| GraphACL | $27.45 \pm 6.04$ | $27.03 \pm 6.04$ | $38.14 \pm 5.59$ | $57.10 \pm 4.72$ | $45.04 \pm 3.37$ | $46.23 \pm 8.33$ | $66.85 \pm 3.98$ | $37.84 \pm 9.56$ |
| **Prompt-based Adaptation** | | | | | | | | |
| GPPT | $28.63 \pm 3.28$ | $35.68 \pm 9.63$ | $43.66 \pm 6.31$ | $39.22 \pm 7.75$ | $38.06 \pm 10.37$ | $49.70 \pm 6.78$ | $56.04 \pm 8.56$ | $39.46 \pm 15.94$ |
| GPF | $34.12 \pm 10.35$ | $28.65 \pm 5.01$ | $33.24 \pm 6.79$ | $46.22 \pm 4.83$ | $34.80 \pm 14.11$ | $42.67 \pm 7.08$ | $60.57 \pm 8.19$ | $41.62 \pm 3.24$ |
| GraphPrompt | $29.02 \pm 6.71$ | $32.43 \pm 10.81$ | $28.94 \pm 2.62$ | $44.42 \pm 5.20$ | $51.44 \pm 3.68$ | $42.62 \pm 7.13$ | $57.73 \pm 7.06$ | $30.27 \pm 5.86$ |
| GraphPrompt+ | $27.06 \pm 4.00$ | $31.35 \pm 11.79$ | $35.14 \pm 5.81$ | $46.12 \pm 5.10$ | $52.16 \pm 4.24$ | $54.43 \pm 4.53$ | $64.42 \pm 7.58$ | $34.59 \pm 9.58$ |
| **Graph Foundation Models** | | | | | | | | |
| GCOPE | $38.21 \pm 4.22$ | $23.44 \pm 12.42$ | $37.52 \pm 1.48$ | $58.71 \pm 1.96$ | $55.26 \pm 1.93$ | $53.01 \pm 3.31$ | $68.15 \pm 2.54$ | $34.90 \pm 11.04$ |
| SAMGPT | $30.98 \pm 7.78$ | $29.73 \pm 11.21$ | $46.46 \pm 10.02$ | $45.76 \pm 3.84$ | $48.26 \pm 6.58$ | $\underline{55.27 \pm 6.04}$ | $62.98 \pm 8.16$ | $25.95 \pm 15.06$ |
| MDGFM | $34.90 \pm 6.37$ | $29.73 \pm 10.40$ | $\underline{47.72 \pm 4.63}$ | $57.52 \pm 6.12$ | $42.86 \pm 4.88$ | $48.44 \pm 10.53$ | $62.84 \pm 4.87$ | $35.14 \pm 11.72$ |
| RiemannGFM | $\underline{40.40 \pm 5.71}$ | $\underline{44.86 \pm 8.65}$ | $47.44 \pm 7.81$ | $58.20 \pm 2.10$ | $53.60 \pm 13.59$ | $40.77 \pm 6.98$ | $\underline{70.82 \pm 3.10}$ | $33.51 \pm 11.29$ |
| R-GFM | $\mathbf{43.10 \pm 4.03}$ | $\mathbf{45.39 \pm 3.09}$ | $\mathbf{73.98 \pm 1.42}$ | $\mathbf{59.26 \pm 5.04}$ | $\mathbf{59.19 \pm 3.01}$ | $\mathbf{56.02 \pm 3.90}$ | $\mathbf{71.50 \pm 2.80}$ | $\mathbf{44.18 \pm 7.02}$ |

Results are reported in percent. The best method is bolded, and the runner-up is underlined.

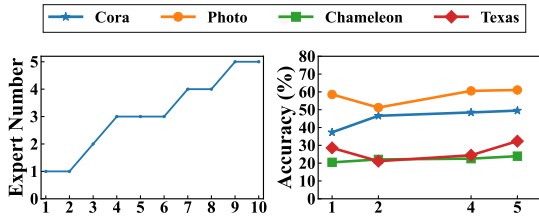

*(a)* Experts $\mathcal{M}$ vs. Datasets $|\mathcal{D}|$. *(b)* Accuracy vs. Experts $\mathcal{M}$.

*Figure 7.* Effect of dataset scale and expert number.

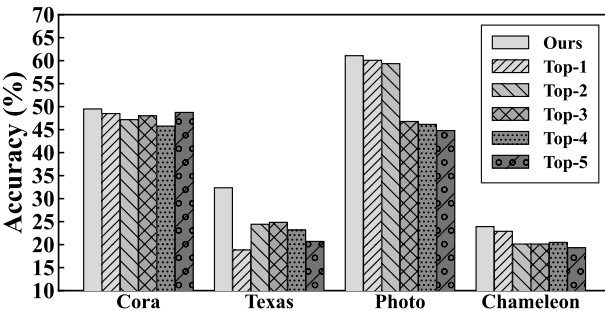

*Figure 8.* Effect of gate sparsity (Top-$m$) across datasets.

edges. We therefore control the edge budget by fixing the retained-edge ratio $\beta \in [0,1]$ ($\beta = \frac{2\mathcal{B}_{edge}}{K(K-1)}$). In Figure 6a, performance peaks around $\beta \approx 0.6$. Therefore, we $\mathcal{B}_{\text{edge}} = 0.6 \times \frac{K(K-1)}{2}$. More analysis is in Appendix H.

**Candidate Expert Number** $\mathcal{M}$. Figure 7a shows that our dynamic selection strategy increases the candidate expert number as more datasets are introduced (added in the order:

Wisconsin, Texas, Cornell, Cora, Citeseer, Pubmed, Computers, Photo, Chameleon, Squirrel), reflecting the growing diversity across tasks. In the full 10-dataset setting, we further vary $\mathcal{M}$ and observe the accuracy trends in Figure 7b; the results indicate that the best-performing $\mathcal{M}$ is dataset-dependent, and our strategy can automatically choose an appropriate expert set without manual tuning.

**Dynamic Gate Sparsity (Top-$m$).** We compare the our dynamic top-$m$ strategy with fixed Top-$m$ gating ($m \in \{1, \dots, 5\}$). In Figure 8, our method achieves the best accuracy across datasets, while fixed $m$ is more sensitive and no single choice generalizes well.

**Temperature $\tau$.** Figure 6b sweeps the temperature in the contrastive objective. We observe a reasonably stable region (e.g., $\tau \in [0.1, 0.5]$), whereas overly small or overly large values can hurt performance on some datasets. We set $\tau = 0.2$ by default, which is competitive across datasets.

## 5. Related Work

One prevalent category of GFMs utilizes GNNs as their primary encoders, including masked graph modeling and contrastive pretraining methods (e.g., DGI (Velickovic et al., 2019), GraphCL (You et al., 2020), and GraphMAE (Hou et al., 2022)). More recent GNN-based GFM designs explicitly address distribution shift across graph domains by topology or structure alignment, prompt-based adaptation, or domain-aware pretraining (e.g., GCOPE (Zhao et al., 2024a), RiemannGFM (Sun et al., 2025), SAMGPT (Yu et al., 2025a)). However, these methods typically use fixed-

*Table 3.* Accuracy evaluation on 5-shot node classification

| Methods | Wisconsin | Cornell | Citeseer | Cora | Pubmed | Computers | Photos | Texas |
|---|---|---|---|---|---|---|---|---|
| **Task-Supervised GNNs** | | | | | | | | |
| GCN | 27.00 ± 6.60 | 20.92 ± 6.02 | 31.61 ± 6.05 | 39.48 ± 6.08 | 38.82 ± 2.98 | 30.06 ± 10.72 | 66.96 ± 9.21 | 38.08 ± 13.65 |
| GAT | 28.85 ± 8.91 | 18.30 ± 7.49 | 23.99 ± 4.10 | 29.82 ± 2.31 | 34.92 ± 12.27 | 37.12 ± 8.21 | 54.37 ± 4.90 | 28.21 ± 11.78 |
| H2GCN | 28.39 ± 13.19 | 35.82 ± 10.30 | 22.29 ± 3.74 | 20.83 ± 6.56 | 42.61 ± 3.61 | 19.67 ± 11.45 | 47.07 ± 3.78 | 32.82 ± 14.56 |
| FAGCN | 24.61 ± 4.70 | 25.36 ± 9.00 | 18.69 ± 1.31 | 15.82 ± 3.01 | 35.92 ± 7.71 | 28.68 ± 1.89 | 50.83 ± 3.79 | 46.15 ± 9.94 |
| **Self-Supervised Pretraining with Fine-tuning** | | | | | | | | |
| DGI | 39.61 ± 3.77 | 38.92 ± 6.78 | 53.66 ± 6.21 | 63.72 ± 2.23 | 57.02 ± 3.47 | 52.03 ± 5.29 | 69.74 ± 4.95 | 42.70 ± 9.04 |
| GraphCL | 39.61 ± 8.25 | 42.70 ± 5.54 | 39.26 ± 3.63 | 49.34 ± 4.94 | 44.88 ± 2.40 | 57.78 ± 3.96 | 69.39 ± 6.46 | 47.03 ± 13.05 |
| GraphACL | 30.98 ± 6.56 | 29.73 ± 1.91 | 44.68 ± 6.36 | 62.90 ± 5.69 | 50.10 ± 5.39 | 51.60 ± 6.46 | 65.22 ± 8.34 | 34.59 ± 7.74 |
| **Prompt-based Adaptation** | | | | | | | | |
| GPPT | 35.29 ± 7.72 | 44.86 ± 4.52 | 45.98 ± 5.97 | 48.78 ± 5.49 | 48.88 ± 8.25 | 30.14 ± 12.77 | 64.89 ± 2.38 | 48.11 ± 10.71 |
| GPF | 33.73 ± 3.37 | 33.51 ± 6.96 | 38.34 ± 6.35 | 49.98 ± 5.98 | 43.00 ± 3.48 | 47.46 ± 6.54 | 70.89 ± 3.81 | 30.81 ± 12.86 |
| GraphPrompt | 31.37 ± 3.10 | 35.14 ± 8.55 | 35.96 ± 1.97 | 48.24 ± 3.43 | 58.56 ± 2.40 | 48.43 ± 4.50 | 64.34 ± 3.99 | 27.03 ± 3.82 |
| GraphPrompt+ | 27.84 ± 5.46 | 32.43 ± 10.11 | 41.60 ± 5.29 | 51.62 ± 2.00 | 58.76 ± 6.22 | 57.04 ± 5.29 | 69.34 ± 2.48 | 45.95 ± 10.81 |
| **Graph Foundation Models** | | | | | | | | |
| GCOPE | 34.38 ± 3.93 | 35.92 ± 9.30 | 47.32 ± 1.59 | 57.33 ± 2.08 | 60.17 ± 1.81 | 55.76 ± 3.86 | 68.28 ± 2.45 | 26.34 ± 10.63 |
| SAMGPT | 39.22 ± 9.20 | 41.08 ± 11.13 | 59.44 ± 5.70 | 62.76 ± 3.93 | 47.08 ± 9.10 | 58.81 ± 7.14 | 72.08 ± 4.20 | 31.35 ± 6.07 |
| MDGFM | 36.08 ± 7.40 | 41.08 ± 7.72 | 57.08 ± 3.71 | 61.86 ± 4.40 | 48.84 ± 9.86 | 49.68 ± 4.93 | 70.37 ± 7.36 | 21.08 ± 5.77 |
| RiemannGFM | 40.40 ± 8.71 | 44.86 ± 4.05 | 54.68 ± 3.67 | 62.92 ± 5.67 | 61.48 ± 6.54 | 48.93 ± 5.75 | 73.47 ± 6.42 | 41.08 ± 12.01 |
| R-GFM | **47.75 ± 4.97** | **47.97 ± 2.82** | **74.59 ± 1.87** | **63.77 ± 2.47** | **63.39 ± 1.85** | **59.55 ± 4.17** | **73.62 ± 1.15** | **51.64 ± 6.31** |

Results are reported in percent. The best method is bolded, and the runner-up is underlined.

*Table 4.* Cross-domain node classification under the LM unified-feature setting.

| Methods | Books-History | Ele-Computers | Cora | Instagram |
|---|---|---|---|---|
| GCOPE | 30.67 ± 7.90 | 23.10 ± 3.38 | 37.80 ± 2.58 | 59.17 ± 7.40 |
| SAMGPT | OOM | OOM | OOM | OOM |
| MDGFM | OOM | OOM | OOM | OOM |
| RiemannGFM | 17.66 ± 9.12 | 24.56 ± 8.97 | 50.92 ± 3.58 | 46.60 ± 9.87 |
| R-GFM | **37.05 ± 11.06** | **30.73 ± 5.31** | **61.78 ± 7.77** | **59.49 ± 4.46** |

*Table 5.* Link prediction performance on representative datasets (**AUC-ROC**, %).

| Methods | Cora | Pubmed | Photos | Texas |
|---|---|---|---|---|
| GCOPE | 88.71 ± 0.57 | 84.99 ± 0.06 | 78.45 ± 5.51 | 80.51 ± 1.75 |
| SAMGPT | 88.62 ± 3.51 | 84.92 ± 0.40 | 80.59 ± 0.33 | 51.20 ± 4.42 |
| MDGFM | 88.41 ± 2.12 | 84.78 ± 2.09 | 81.24 ± 5.80 | 65.22 ± 4.05 |
| RiemannGFM | 88.54 ± 0.11 | 86.37 ± 0.33 | 81.44 ± 0.95 | 72.41 ± 5.96 |
| R-GFM | **89.27 ± 0.64** | **88.66 ± 5.16** | **81.53 ± 0.83** | **87.94 ± 0.96** |

hop sampling, which limits the ability to match the multi-scale structural information required by different tasks.

A natural way to model multi-scale graph structure is to construct a Graph-of-Graphs (GoG), where each node in the higher-level graph corresponds to a graph or subgraph instance. However, existing GoG methods are mainly designed for static graph collections and are therefore not directly applicable to cross-domain graph foundation model pretraining. For example, G2GNN (Wang et al., 2020a) constructs the GoG according to kernel similarity between graph instances, while ImbGNN (Xu et al., 2024) adopts similarity-aware random walks to extract informative sub-

graphs, which can introduce considerable computational overhead. In contrast, our GoG construction is dynamic and tailored to graph foundation models.

Another line integrates GNN encoders with LLMs. For example, OFA (Liu et al., 2024) trains a single model across domains and tasks via a unified prompting paradigm and language-based descriptions to bridge heterogeneous graph feature spaces. GraphGPT (Tang et al., 2024) aligns LLMs with graph structural knowledge through graph instruction tuning and lightweight graph-text projection modules. Such models treat the GNN encoder as a modular component, and performance depends critically on how well the GNN module captures structure information (Kong et al., 2025).

R-GFM complements these GNN+LLM-based GFMs, as it can be seamlessly integrated as an advanced GNN encoder. This integration bolsters structural modeling capabilities while preserving the semantic reasoning abilities of LLMs.

# 6. Conclusion

We study graph foundation models in the challenging cross-domain setting, where graphs exhibit substantial heterogeneity in topology and geometry. To improve transferability, we propose R-GFM, a Riemannian Graph-of-Graphs GFM framework that (1) constructs Graph-of-Graphs by adaptive-hop subgraphs based on similarity, and (2) adapts to diverse topologies via dynamic routing to a mixture of Riemannian experts. Empirically, R-GFM consistently improves cross-domain generalization across diverse benchmarks, highlighting that both geometry-aware specialization and multi-scale structural modeling are essential for robust graph transfer.

## Impact Statement

Our work presents a new graph foundation model that aims to advance the field of Machine Learning, particularly graph machine learning. This work is primarily foundational research. While our work may be applied in many real-world settings, we do not foresee any immediate negative societal impacts that are specific to this work.

## Acknowledgements

This work was supported by the National Natural Science Foundation of China (NSFC) under Grant 62472400.

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

## Appendix Overview

In the Appendix, we provide additional details organized as follows:

## A. Notations

*Table 6.* Notations.

| Notation | Description |
|---|---|
| $G = (V, E, X_V)$ | An attributed graph with node set $V$, edge set $E$, and node features $X_V$. |
| $\{G_i\}$ | A set/pool of graphs from different domains. |
| $D_i$ | The domain associated with graph/dataset $i$. |
| $D = \{D_i\}$ | The set of all domains. |
| $\mathrm{dist}(u, v)$ | Shortest-path distance between nodes $u$ and $v$ in $G$. |
| $N_k(u)$ | $k$-hop neighbors of node $u$: $N_k(u) = \{v \in V : \mathrm{dist}(u, v) \leq k\}$. |
| $G_{\mathrm{GoG}} = (V_{\mathrm{GoG}}, E_{\mathrm{GoG}}, X_G)$ | Graph-of-graphs (GoG) with GoG nodes $V_{\mathrm{GoG}}$, edges $E_{\mathrm{GoG}}$, and features $X_G$. |
| $v$ | A center (training) node in the original graph, used to construct a center-local GoG. |
| $\{G_v^{(i)}\}_{i=1}^{K}$ | Adaptive-hop sampled subgraphs (GoG nodes) from hop 1 to hop $K$ around center $v$. |
| $K$ | Hop number; also equals the number of GoG nodes for each center-local GoG. |
| $B_{\mathrm{GPU}}$ | GPU memory budget used to determine the maximal feasible $K$. |
| $X_{\mathrm{sub}} \in \mathbb{R}^{K \times d}$ | Subgraph embedding matrix (each row is a subgraph embedding). |
| $S \in \mathbb{R}^{K \times K}$ | Subgraph similarity matrix (cosine; implemented as $S = X_{\mathrm{sub}} X_{\mathrm{sub}}^{\top}$). |
| $\mathrm{Prob}(i, j)$ | Sampling probability of GoG edge $(i, j)$ after normalizing similarity scores. |

| Notation | Description |
|---|---|
| $B_{\text{edge}}$ | Edge budget: number of GoG edges to sample (before symmetrization). |
| $E_{\text{GoG}}$ | Sampled GoG edge set under budget $B_{\text{edge}}$. |
| $e$ | Euler's number (used in the normalization for $\text{Prob}$). |
| $x$ | (Random) subgraph embedding viewed as a noisy variable. |
| $\mu$ | Noise-free subgraph embedding (mean of $x$). |
| $\sigma$ | Embedding noise (std); smaller $\sigma$ indicates higher quality. |
| $\sigma_F$ | Noise when sampling from a fixed hop. |
| $\sigma_V$ | Noise when sampling from various hops (adaptive-hop strategy). |
| $\|\cdot\|_2$ | $\ell_2$ norm. |
| $e_{none}$ | Expected squared embedding error under a GoG without edges. |
| $e_w$ | Expected squared embedding error with GoG edges. |
| $e_{full}$ | Expected squared embedding error under a fully connected GoG. |
| $\hat{\mu}_{none}$ | Subgraph-embedding estimators with constructed a GoG without edges. |
| $\hat{\mu}_w$ | Subgraph-embedding estimators with constructed GoG edges. |
| $\hat{\mu}_{full}$ | Subgraph-embedding estimators with fully connected GoG edges. |
| $\kappa$ | Constant curvature of a manifold (e.g., $\kappa < 0$ Hyperbolic, $\kappa = 0$ Euclidean, $\kappa > 0$ Hyperspherical). |
| $\mathcal{K}$ | Candidate curvature set (selected based on structure statistics). |
| $\mathcal{M}$ | Candidate Riemannian expert set induced by $\mathcal{K}$. |
| $\psi$ | Number of candidate experts (size of $\mathcal{M}$). |
| $\deg(D_i)$ | Node-degree values of dataset/graph $D_i$. |
| $\text{CV}(D_i)$ | Coefficient of variation of degree distribution: $\text{std}(\deg(D_i))/\text{mean}(\deg(D_i))$. |
| $c$ | Center index (each center induces a GoG). |
| $k \in \{1, \ldots, K\}$ | GoG node index (corresponding to hop $k$). |
| $j \in \{1, \ldots, \psi\}$ | Expert index. |
| $z_{c,k}^{(j)}$ | Router logits for assigning GoG node $(c, k)$ to expert $j$. |
| $\alpha_{c,k}^{(j)}$ | Routing weights (softmax over experts) for GoG node $(c, k)$. |
| $\tau$ | Temperature in the softmax routing distribution. |
| $\text{conf}_{c,k}$ | Confidence score, defined as $\max_j \alpha_{c,k}^{(j)}$. |
| $m$ | Number of activated experts in Top-$m$ routing (dynamic). |
| $\odot$ | Element-wise multiplication (masking non-Top-$m$ experts). |
| $\mathbf{1}[\cdot]$ | Indicator function (e.g., $\mathbf{1}[\text{rank}(\alpha) \leq m]$). |
| $\tilde{\alpha}_{c,k}^{(j)}$ | Masked-and-renormalized routing weights after Top-$m$. |
| $u_j$ | Normalized usage / selection frequency of expert $j$ (for balancing). |
| $\mathcal{L}_{\text{lb}}$ | Load-balancing loss to mitigate expert collapse. |

# B. Time Complexity Analysis

---

**Algorithm 1** Dynamical Candidate Expert set Determination

---

**Input:** dataset pool $\{\mathcal{D}_1, \ldots, \mathcal{D}_i\}$ with full graphs $\{G_{\mathcal{D}_j}\}_{j=1}^i$ (self-loops added), scale $\zeta$, predefined ordered curvature list $\mathcal{K}_0$

**Output:** candidate curvature set $\mathcal{K}_{\mathcal{D}_i}$, candidate expert set $\mathcal{M}_{\mathcal{D}_i}$, expert count $M_{\mathcal{D}_i}$

**for** $j = 1$ **to** $i$ **do**

  compute $CV(\mathcal{D}_j) = \frac{\text{std}(\deg(\mathcal{D}_j))}{\text{mean}(\deg(\mathcal{D}_j))}$                (Eq. (5))

**end for**

$\mu_i \leftarrow \text{mean}(\{CV(\mathcal{D}_1), \ldots, CV(\mathcal{D}_i)\}), \ \sigma_i \leftarrow \text{std}(\{CV(\mathcal{D}_1), \ldots, CV(\mathcal{D}_i)\})$

$S_i \leftarrow \text{normalize}(\mu_i + \sigma_i)$                    (Eq. (6))

$M_{\mathcal{D}_i} \leftarrow \min(|\mathcal{K}_0|, \lceil S_i \cdot \zeta \rceil)$

$\mathcal{K}_{\mathcal{D}_i} \leftarrow \mathcal{K}_0[1 : M_{\mathcal{D}_i}]$                  (prefix of ordered list)

$\mathcal{M}_{\mathcal{D}_i} \leftarrow \{\text{GNN}_\kappa \mid \kappa \in \mathcal{K}_{\mathcal{D}_i}\}$

**return** $\mathcal{K}_{\mathcal{D}_i}, \mathcal{M}_{\mathcal{D}_i}, M_{\mathcal{D}_i}$

---

---

**Algorithm 2** Subgraph Similarity-based GoG Edge Construction

---

**Input:** hop-wise embeddings $\{\mathbf{x}_{c,k}\}_{k=1}^K$, edge budget $\mathcal{B}_{edge}$

**Output:** center-local GoG $\mathcal{G}_c = (\mathcal{V}_c, \mathcal{E}_c)$ and node features $\mathbf{X}_c$

$\mathcal{V}_c \leftarrow \{1, \ldots, K\}$

$\mathbf{X}_c \leftarrow [\mathbf{x}_{c,1}; \ldots; \mathbf{x}_{c,K}] \in \mathbb{R}^{K \times d}$

row-normalize $\mathbf{X}_c$ to obtain $\bar{\mathbf{X}}_c$

$\mathbf{S}_c \leftarrow \bar{\mathbf{X}}_c(\bar{\mathbf{X}}_c)^\top \in \mathbb{R}^{K \times K}$                (Eq. (1))

$\mathcal{U}_c \leftarrow \{(i,j) \mid 1 \le i < j \le K\}$

$q_c(i,j) \leftarrow \frac{e^{\mathbf{S}[i,j]}}{\sum_u \sum_v e^{\mathbf{S}[u,v]}}, \ \forall (i,j) \in \mathcal{U}_c$         (Eq. (2))

sample $\tilde{\mathcal{E}}_c \leftarrow \text{SampleWOR}(q_c; B_e^{\text{gog}})$           (Eq. (3))

$\mathcal{E}_c \leftarrow \{(i,j), (j,i) \mid (i,j) \in \tilde{\mathcal{E}}_c\}$

**return** $\mathcal{G}_c = (\mathcal{V}_c, \mathcal{E}_c)$ and $\mathbf{X}_c$

---

---

**Algorithm 3** Confidence-aware Dynamic Top-$m$ Routing

---

**Input:** GoG $\mathcal{G}_c = (\mathcal{V}_c, \mathcal{E}_c)$, initial hop features $\mathbf{X}_c = \{\mathbf{x}_{c,k}\}_{k=1}^K$, candidate experts $\mathcal{M}_\mathcal{D} = \{\text{GNN}_\kappa \mid \kappa \in \mathcal{K}_\mathcal{D}\}$ with $M_\mathcal{D} = |\mathcal{K}_\mathcal{D}|$, shortlist $m_{\text{start}}$, active count $m$, minimum $m_{\min}$, temperature $\tau$

**Output:** refined hop embeddings $\{\mathbf{h}_{c,k}\}_{k=1}^K$, load-balancing loss $\mathcal{L}_{\text{lb}}$

**for** $k = 1$ **to** $K$ **do**

  $\mathbf{z}_{c,k} \leftarrow g(\mathcal{G}_c, \mathbf{x}_{c,k}) \in \mathbb{R}^{M_\mathcal{D}}$               (router logits)

  $\boldsymbol{\alpha}_{c,k} \leftarrow \text{softmax}(\mathbf{z}_{c,k}/\tau) \in \mathbb{R}^{M_\mathcal{D}}$

**end for**

$\bar{\boldsymbol{\alpha}}_c \leftarrow \frac{1}{K} \sum_{k=1}^K \boldsymbol{\alpha}_{c,k}$

$\mathcal{I}_c^{\text{top}} \leftarrow \text{TopK}(\bar{\boldsymbol{\alpha}}_c, m_{\text{start}})$             (indices in $[1, M_\mathcal{D}]$)

$\tilde{\boldsymbol{\alpha}}_c \leftarrow \text{Normalize}\Big(\text{softmax}(\bar{\boldsymbol{\alpha}}_c|_{\mathcal{I}_c^{\text{top}}}) \odot \mathbf{1}[\text{rank} \le m]\Big)$

**compute only shortlisted experts:** $\mathbf{H}_c^{(j)} \leftarrow \text{GNN}_{\kappa_j}(\mathcal{G}_c, \mathbf{X}_c)$ for $j \in \mathcal{I}_c^{\text{top}}$

**for** $k = 1$ **to** $K$ **do**

  $\mathbf{h}_{c,k} \leftarrow \sum_{j \in \mathcal{I}_c^{\text{top}}} \tilde{\alpha}_c^{(j)} \mathbf{H}_c^{(j)}[k]$

**end for**

$u_j \leftarrow \mathbb{E}_{c,k}[\alpha_{c,k}^{(j)}], \ \mathcal{L}_{\text{lb}} \leftarrow \|\mathbf{u} - \frac{1}{M_\mathcal{D}}\mathbf{1}\|_2^2$

update $m \leftarrow \max(m_{\min}, m - \Delta(\text{conf}))$

**return** $\{\mathbf{h}_{c,k}\}_{k=1}^K, \mathcal{L}_{\text{lb}}$

---

**Dynamical Candidate Expert set Determination (Algorithm 1).** Computing node degrees on the full graph $G_{\mathcal{D}} = (V_{\mathcal{D}}, E_{\mathcal{D}})$ takes $\mathcal{O}(|E_{\mathcal{D}}|)$, and computing the coefficient of variation (CV) and score statistics over degrees takes $\mathcal{O}(|V_{\mathcal{D}}|)$. Mapping the score to $M_{\mathcal{D}}$ and prefix truncation are $\mathcal{O}(M_{\mathcal{D}})$. Overall, the candidate-set determination cost is

$$\mathcal{O}\big(|E_{\mathcal{D}}| + |V_{\mathcal{D}}|\big), \tag{11}$$

and is incurred once per dataset (amortized across all training steps).

**Subgraph Similarity-based GoG Edge Construction (Algorithm. 2).** Given hop vectors, row-normalization costs $\mathcal{O}(Kd)$. The similarity matrix $\mathbf{S} = \mathbf{X}(\bar{\mathbf{X}})^{\top}$ costs

$$\mathcal{O}(K^2 d). \tag{12}$$

Forming the pair set $\mathcal{U}_c$ and computing the sampling distribution $q_c(i, j)$ over $\Theta(K^2)$ unordered pairs costs $\mathcal{O}(K^2)$, and sampling $\mathcal{B}_{edge}$ edges without replacement can be implemented in $\tilde{\mathcal{O}}(K^2)$ time. After symmetrization, $E_c = 2\mathcal{B}_{edge}$.

**Confidence-aware Dynamic Top-$m$ Routing (Algorithm 3).** The router is a 2-layer GCN on $\mathcal{G}_c$. Using the standard decomposition of (i) sparse propagation and (ii) linear transforms, each layer costs $\mathcal{O}(E_c d_r + K d_r^2)$, hence the router costs

$$\mathcal{O}\big(E_c d_r + K d_r^2\big). \tag{13}$$

Computing the shortlist via $\mathrm{TopK}(\cdot, m_{\text{start}})$ over $M_{\mathcal{D}}$ experts costs $\mathcal{O}(M_{\mathcal{D}} \log m_{\text{start}})$ per center (or $\mathcal{O}(M_{\mathcal{D}})$ with selection algorithms). The remaining normalization and masking are $\mathcal{O}(M_{\mathcal{D}})$.

**Riemannian expert computation and fusion.** Assume each activated expert performs $L$ layers of curvature-aware message passing on the same GoG topology. Per expert, the cost is $\mathcal{O}\big(L(E_c d' + K d'^2)\big)$, hence activating only $m$ experts yields

$$\mathcal{O}\big(m\, L(E_c d' + K d'^2)\big) \ \leq\ \mathcal{O}\big(m_{\text{start}}\, L(E_c d' + K d'^2)\big), \tag{14}$$

instead of $\mathcal{O}\big(M_{\mathcal{D}}\, L(E_c d' + K d'^2)\big)$ under dense routing. The final fusion $\sum_m \tilde{\alpha}_c^{(m)} \mathbf{H}_c^{(m)}[k]$ costs $\mathcal{O}(mKd')$.

**End-to-end per-center complexity.** Combining the above, the dominant per-center cost is

$$\mathcal{O}\big(K^2 d\big) \ +\ \mathcal{O}\big(E_c d_r + K d_r^2\big) \ +\ \mathcal{O}\big(m\, L(E_c d' + K d'^2)\big), \tag{15}$$

where $K$ is explicitly controlled by the GPU memory budget and $E_c = 2\mathcal{B}_{edge}$ by the GoG edge budget.

## C. Dataset Detail

### C.1. Main setting: 10 benchmark graphs

**Benchmarks.** Our main setting uses 10 benchmark graphs covering four domains: citations (Cora, CiteSeer, PubMed), webpage hyperlinks (Cornell, Texas, Wisconsin), Wikipedia page networks (Chameleon, Squirrel), and e-commerce co-purchase graphs (Amazon-Computers, Amazon-Photo) (Sen et al., 2008; Yang et al., 2016; Pei et al., 2020; McAuley et al., 2015; Shchur et al., 2018). We follow a leave-one-dataset-out protocol on this benchmark suite: for each target dataset, we pretrain on the remaining datasets and then evaluate transfer on the held-out target. Table 7 summarizes dataset statistics for the main setting.

**Why exclude Chameleon and Squirrel as targets.** Recent work reports severe issues in Chameleon and Squirrel (e.g., duplicate nodes that may induce train-test leakage), which can make cross-method comparisons unreliable (Platonov et al., 2023). To ensure fair comparisons while preserving domain coverage, we still include Chameleon and Squirrel as pretraining sources in this main setting, but we exclude them from downstream evaluation when reporting the main transfer results (i.e., we report results on the remaining 8 target datasets). In later experiments, we additionally use Chameleon as a targeted downstream testbed to verify effectiveness on the Wikipedia-network domain.

*Table 7.* Benchmark datasets used in standard evaluation.

| Dataset | #Nodes | #Edges | Class | Domain |
|---|---|---|---|---|
| Cora | 2,708 | 5,278 | 7 | Academic |
| CiteSeer | 3,327 | 4,552 | 6 | Academic |
| PubMed | 19,717 | 44,324 | 3 | Academic |
| Cornell | 183 | 277 | 5 | Webpage |
| Texas | 183 | 279 | 5 | Webpage |
| Wisconsin | 251 | 450 | 5 | Webpage |
| Chameleon | 2,277 | 31,371 | 5 | Wikipedia |
| Squirrel | 5,201 | 198,353 | 5 | Wikipedia |
| Amazon-Computers | 13,752 | 245,861 | 10 | E-commerce |
| Amazon-Photo | 7,650 | 119,081 | 8 | E-commerce |

*Table 8.* LM-text scaling setting: dataset statistics for source and target graphs.

| | #Nodes | #Edges | Domain | Class | Usage |
|---|---|---|---|---|---|
| ogbn-ArXiv | 169,343 | 2,315,598 | Academic | 40 | $G_{\text{source}}$ |
| ArXiv_2023 | 33,868 | 611,344 | Academic | 40 | $G_{\text{source}}$ |
| PubMed | 19,717 | 44,324 | Academic | 3 | $G_{\text{source}}$ |
| Reddit | 33,434 | 269,442 | Social | 2 | $G_{\text{source}}$ |
| Cora | 2,708 | 5,278 | Academic | 7 | $G_{\text{target}}$ |
| Ele-Computers | 87,229 | 1,256,548 | E-commerce | 10 | $G_{\text{target}}$ |
| Books-History | 41,551 | 503,180 | E-commerce | 12 | $G_{\text{target}}$ |
| Instagram | 11,339 | 144,010 | Social | 2 | $G_{\text{target}}$ |

### C.2. Large-scale setting: unified semantic node features and large graphs

**Motivation.** To validate R-GFM under (i) a unified semantic feature space across heterogeneous graphs and (ii) larger-scale real-world graphs, we consider an LM-text scaling setting.

**Text-based node features.** For datasets with raw node texts, we encode each node's text using a frozen language model (Sentence-BERT all-MiniLM-L6-v2 (Reimers & Gurevych, 2020)) and use the resulting embeddings as node features. This provides semantically rich and fixed-dimensional representations shared across datasets.

**Pretraining and evaluation.** Under this setting, we pretrain on ArXiv_2023 (He et al., 2023), ogbn-Arxiv (Wang et al., 2020b), Reddit (Huang et al., 2024b), and PubMed (Yang et al., 2016), and evaluate transfer on Cora (Sen et al., 2008), Ele-Computers and Books-History (Yan et al., 2023), and Instagram (Huang et al., 2024b). Table 8 reports detailed statistics for the LM-text scaling setting.

### C.3. Datasets description

**Citation networks.** We use several citation graphs including Cora and CiteSeer (Sen et al., 2008), PubMed (Yang et al., 2016), ogbn-ArXiv (Wang et al., 2020b), and ArXiv_2023 (He et al., 2023). In these datasets, nodes represent papers and edges indicate citation links. Each node is associated with textual attributes of the paper (e.g., title and abstract), and a bag-of-words feature vector depending on the benchmark construction. The node label corresponds to the topic or category of the paper. Compared with Cora and CiteSeer that cover diverse computer science topics, PubMed consists of

*Table 9.* Link prediction results (**AUC-ROC**(%)). Results are reported in percent. The best method is bolded and the runner-up is underlined.

| Methods | Wisconsin | Cornell | Citeseer | Pubmed | Cora | Photos | Texas |
|---|---|---|---|---|---|---|---|
| GCOPE | $83.12 \pm 1.51$ | $64.72 \pm 4.99$ | $85.09 \pm 0.34$ | $84.99 \pm 0.06$ | $88.71 \pm 0.57$ | $78.45 \pm 5.51$ | $80.51 \pm 1.75$ |
| SAMGPT | $65.32 \pm 7.27$ | $53.53 \pm 4.57$ | $89.91 \pm 2.70$ | $84.92 \pm 0.40$ | $88.62 \pm 3.51$ | $80.59 \pm 0.33$ | $51.20 \pm 4.42$ |
| MDGFM | $67.46 \pm 6.82$ | $54.92 \pm 21.10$ | $89.12 \pm 4.68$ | $84.78 \pm 2.09$ | $88.41 \pm 2.12$ | $81.24 \pm 5.80$ | $65.22 \pm 4.05$ |
| RiemannGFM | $78.19 \pm 2.08$ | $\underline{79.06 \pm 2.96}$ | $\underline{90.63 \pm 0.58}$ | $\underline{86.37 \pm 0.33}$ | $88.54 \pm 0.11$ | $\underline{81.44 \pm 0.95}$ | $72.41 \pm 5.96$ |
| R-GFM | $\mathbf{84.15 \pm 0.71}$ | $\mathbf{85.90 \pm 0.69}$ | $\mathbf{90.88 \pm 0.70}$ | $\mathbf{88.62 \pm 0.41}$ | $\mathbf{89.27 \pm 0.64}$ | $\mathbf{81.53 \pm 0.83}$ | $\mathbf{87.94 \pm 0.96}$ |

biomedical publications focused on diabetes, where labels indicate the type of diabetes studied (Yang et al., 2016). For ogbn-ArXiv (Wang et al., 2020b) and ArXiv_2023 (He et al., 2023), labels follow arXiv subject areas with 40 classes.

**Webpage hyperlink networks (WebKB).** We evaluate on the WebKB benchmarks Cornell, Texas, and Wisconsin (Pei et al., 2020), which are subsets derived from the WebKB corpus. Each node corresponds to a web page, and edges denote hyperlinks between pages. Node features are bag-of-words representations extracted from page content. Nodes are classified into five categories: student, project, course, staff, and faculty (Pei et al., 2020).

**Wikipedia page networks.** We use Chameleon and Squirrel (Pei et al., 2020), two page-to-page graphs built from Wikipedia. Nodes represent web pages and edges represent page links. Node attributes are constructed from the page content (e.g., collections of nouns/words), and node labels correspond to discretized levels of average monthly traffic of the pages (Pei et al., 2020).

**E-commerce graphs.** We consider multiple product and item graphs, including Amazon-Computers and Amazon-Photo (McAuley et al., 2015; Shchur et al., 2018), as well as Ele-Computers and Books-History (Yan et al., 2023). In these datasets, nodes denote products/items and edges connect related items defined by the benchmark construction (e.g., co-purchase/co-interaction relations). Each node is associated with an attribute vector and/or text fields such as product title/description. Node labels indicate item categories, with 10 classes for Amazon-Computers and Ele-Computers, 8 classes for Amazon-Photo, and 12 classes for Books-History (McAuley et al., 2015; Shchur et al., 2018; Yan et al., 2023).

**Online social networks.** We use Reddit (Huang et al., 2024b) and Instagram (Huang et al., 2024b). These datasets represent large-scale online platforms, where nodes correspond to entities defined by the benchmark (e.g., users or content units) and edges capture their relations/interactions. Nodes are associated with raw text attributes when available, and the tasks are binary node classification with 2 classes (Huang et al., 2024b).

# D. Additional Experiments

## D.1. Additional link prediction results

Table 9 reports the link prediction performance in terms of AUC-ROC. Overall, R-GFM consistently achieves the best results across all evaluated datasets, outperforming prior GFMs by clear margins. Compared with the strongest baseline in each case (typically RiemannGFM), R-GFM brings absolute improvements of +1.03 on Wisconsin, +6.84 on Cornell, +0.25 on Citeseer, +2.25 on Pubmed, +0.56 on Cora, +0.09 on Photos, and +7.43 on Texas AUC-ROC. Notably, the gains are substantially larger on heterophilic WebKB graphs (Wisconsin/Cornell/Texas), indicating that R-GFM is more robust to structural heterogeneity, while still remaining competitive on citation and co-purchase networks (Citeseer/Pubmed/Cora/Photos).

## D.2. Additional scalability results

Figure 9 reports the scalability comparison of different graph foundation models in terms of training time. Overall, R-GFM demonstrates favorable scalability as the graph size increases, maintaining competitive training efficiency compared with existing GFMs. In particular, R-GFM scales more efficiently than GCOPE and RiemannGFM on large-scale graphs, while achieving substantially better predictive performance as shown in the main experiments. These results indicate that R-GFM not only improves effectiveness but also remains practical for large-scale graph applications.

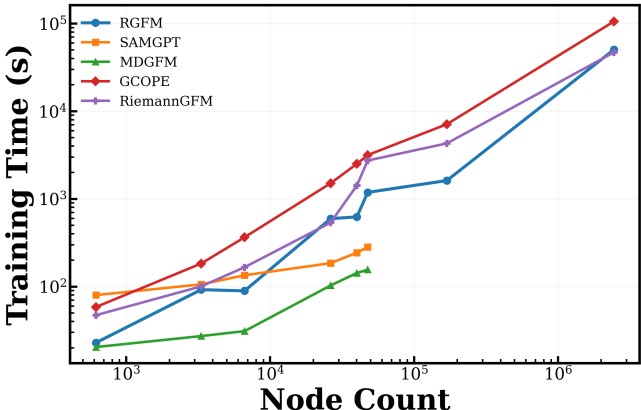

*Figure 9.* Scalability comparison in terms of training time with respect to the number of nodes.

*Table 10.* Transfer pretraining results on graph classification. Models are pretrained on ChEMBL and fine-tuned on HIV. The best result is bolded and the runner-up is underlined.

| Model | Test AUC |
|---|---|
| MDGFM | 0.7563 |
| SAMGPT | 0.7664 |
| RiemannGFM | 0.7804 |
| GCOPE | 0.7230 |
| R-GFM | **0.7849** |

### D.3. Graph Classification

To further evaluate the generalizability of R-GFM beyond node-level tasks, we conduct additional graph classification experiments on HIV (Wu et al., 2017; Hu et al., 2020). Specifically, models are pretrained on the large-scale molecular graph dataset ChEMBL (Gaulton et al., 2012) and then fine-tuned on the downstream HIV graph classification benchmark.

As shown in Tables 10 and 11, R-GFM achieves the best Test AUC in both transfer and few-shot graph classification settings. These results indicate that R-GFM can generalize from node-level tasks to graph-level prediction while maintaining consistent advantages over existing GFM baselines.

## E. Experimental Details

### E.1. Baseline Selection

**Why these baselines.**    We select four categories of baselines to cover the most common training and adaptation paradigms for graph transfer. (1) **Task-supervised GNNs** (GCN (Kipf & Welling, 2016), GAT (Velickovic et al., 2017), H2GCN (Zhu et al., 2020), FAGCN (Bo et al., 2021)) are included as widely-used supervised architectures spanning homophily-oriented and heterophily-aware designs, which is important given the diverse structural properties across our benchmarks. (2) **Self-supervised pretraining with fine-tuning** baselines (DGI (Velickovic et al., 2019), GraphCL (You et al., 2020), GraphACL (Xiao et al., 2023), MixHop (Abu-El-Haija et al., 2019)) represent representative self-supervised objectives for learning transferable node/graph representations before downstream adaptation. (3) **Prompt-based adaptation** baselines (GPPT (Sun et al., 2022), GPF (Fang et al., 2023), GraphPrompt (Liu et al., 2023), GraphPrompt+ (Yu et al., 2024)) are included as parameter-efficient alternatives to full fine-tuning, which is a common setting in recent graph transfer literature. (4) **Existing GFMs** (GCOPE (Zhao et al., 2024a), SAMGPT (Yu et al., 2025a), MDGFM (Wang et al., 2025a), RiemannGFM (Sun et al., 2025), GPM (Wang et al., 2025c), G2PM (Wang et al., 2025d)) are included as recent general-purpose graph foundation models that aim to scale pretraining across diverse graphs and transfer via fine-tuning or lightweight adaptation. Following common practice on graphs with varying homophily/heterophily, we adopt FAGCN as the Stage-1 encoder in R-GFM for its robustness across both regimes.

*Table 11.* Few-shot graph classification results on HIV.

| Method | 1-shot | 3-shot | 5-shot |
|---|---|---|---|
| Multi-RF Fusion | 0.5498 | 0.5764 | 0.5913 |
| DeepAUC | 0.5172 | 0.5910 | 0.6068 |
| HIG | 0.5433 | 0.5913 | 0.5973 |
| PAS-OGB | 0.5003 | 0.5948 | 0.6082 |
| MDGFM | 0.6101 | 0.6169 | 0.6365 |
| SAMGPT | 0.6170 | 0.6248 | 0.6424 |
| RiemannGFM | 0.6175 | 0.6273 | 0.6368 |
| GCOPE | 0.5825 | 0.6354 | 0.6536 |
| R-GFM | **0.6473** | **0.6557** | **0.6648** |

*Table 12.* Settings of transfer evaluation on the 10-benchmark suite.

| Target | Citations | | | Webpage | | | Wikipedia | | E-commerce | |
|---|---|---|---|---|---|---|---|---|---|---|
| | Cora | CiteSeer | PubMed | Cornell | Texas | Wisconsin | Chameleon | Squirrel | Amazon-Computers | Amazon-Photo |
| Cora | | ✓ | ✓ | ✓ | ✓ | ✓ | ✓ | ✓ | ✓ | ✓ |
| CiteSeer | ✓ | | ✓ | ✓ | ✓ | ✓ | ✓ | ✓ | ✓ | ✓ |
| PubMed | ✓ | ✓ | | ✓ | ✓ | ✓ | ✓ | ✓ | ✓ | ✓ |
| Cornell | ✓ | ✓ | ✓ | | ✓ | ✓ | ✓ | ✓ | ✓ | ✓ |
| Texas | ✓ | ✓ | ✓ | ✓ | | ✓ | ✓ | ✓ | ✓ | ✓ |
| Wisconsin | ✓ | ✓ | ✓ | ✓ | ✓ | | ✓ | ✓ | ✓ | ✓ |
| Amazon-Computers | ✓ | ✓ | ✓ | ✓ | ✓ | ✓ | ✓ | ✓ | | ✓ |
| Amazon-Photo | ✓ | ✓ | ✓ | ✓ | ✓ | ✓ | ✓ | ✓ | ✓ | |

## E.2. Expertiment Settings

In the standard benchmark setting, we follow a leave-one-dataset-out protocol: for each target dataset, we pretrain on all remaining datasets and evaluate transfer on the held-out target (Table 12). In the large-scale setting, we pretrain on the designated large graphs and evaluate transfer on the designated targets as summarized in Table 13.

**Hardware and software.** All experiments are conducted on a server with $2\times$ Intel(R) Xeon(R) Platinum 8358 CPU @ 2.60GHz, 377 GiB RAM, and NVIDIA A100 80GB PCIe GPUs (driver 550.54.15); unless otherwise stated, experiments use a single A100 80GB GPU. The software stack (conda environment `prog`) includes Python 3.10.13, PyTorch 2.1.0+cu118, DGL 2.4.0+cu118, PyG 2.6.1, OGB 1.3.6, and Transformers 4.36.2. Unless otherwise stated, each result is averaged over five runs with different random seeds.

**Implementation details.** Both tasks use a per-GPU batch size of 128 and FAGCN (dropout 0.2), and we use temperature $\tau = 0.2$ for training.

For node tasks, training follows two stages: Stage 1 trains the encoder for 100 epochs with Adam (lr $5 \times 10^{-3}$, wd $2 \times 10^{-6}$), and Stage 2 optimizes the MoE for 50 epochs with Adam (lr $10^{-2}$, wd $2 \times 10^{-6}$) together with RiemannianAdam (lr $10^{-3}$, wd $5 \times 10^{-4}$, stabilize=100).

For link tasks, `edge_task.sh` uses two stages with 200 epochs each (total 400 epochs): Stage 1 trains the encoder for 200 epochs and Stage 2 optimizes the MoE for 200 epochs, with `encoder_lr = moe_lr = classifier_lr` $= 3 \times 10^{-3}$, `riemannian_lr` $= 10^{-3}$, wd $10^{-5}$, and load-balance weight 0.01.

## E.3. Code-level Training and Evaluation Pipeline

Our released implementation instantiates the four-stage workflow in Fig. 2 (Stage A–D) as an end-to-end, two-stage routine: (i) Stage-1 subgraph encoder training and (ii) Stage-2 GoG-MoE training, followed by downstream evaluation.

*Table 13.* Large scale setting: transfer protocol.

| Target | Academic sources | | | Social source |
|---|---|---|---|---|
| | ogbn-ArXiv | ArXiv_2023 | PubMed | Reddit |
| Cora | ✓ | ✓ | ✓ | ✓ |
| Ele-Computers | ✓ | ✓ | ✓ | ✓ |
| Books-History | ✓ | ✓ | ✓ | ✓ |
| Instagram | ✓ | ✓ | ✓ | ✓ |

**Entry points.** `main.py` runs node-level tasks and `main-link.py` runs link-level tasks. Each entry script performs data preparation, (sub)graph construction and caching, Stage-1 training, Stage-2 training, evaluation, and checkpointing in one run.

**Step 1: Data acquisition and preprocessing.** For node classification, the loader downloads datasets (PyG/OGB) on demand, standardizes node features by clipping or padding to a fixed dimensionality, and stores `x/y/edge_index` tensors. For link prediction, the loader applies a randomized split (e.g., `RandomLinkSplit` with held-out validation/test edges and negative sampling).

**Step 2: k-hop subgraph construction and caching.** For each training center (node or edge), the implementation constructs $1 \ldots K$ hop ego-subgraphs and caches them to disk. This implements the adaptive-hop idea in the paper: as $K$ increases, the model captures larger structural scales but incurs higher GPU memory cost; therefore, $K$ is treated as a budget-controlled parameter and the implementation includes OOM-safe fallback.

**Step 3: Graph augmentations and GraphCL views (node tasks).** For node tasks, two augmented views are sampled per subgraph to form a GraphCL-style contrastive pair (e.g., node dropping, edge perturbation, and attribute masking with a sampled corruption ratio). Augmented views can be precomputed and cached to reduce repeated preprocessing overhead.

**Step 4: Stage-1 training.** **Node tasks:** train a GNN subgraph encoder with a contrastive objective (GraphCL-style) on sampled subgraph pairs. **Link tasks:** train the encoder (and a lightweight classifier) with supervised edge labels constructed from positive/negative pairs, using k-hop features as input.

**Step 5: Stage-2 GoG-MoE training.** Subgraph embeddings from multiple hops are treated as nodes in a higher-order Graph-of-Graphs (GoG). The GoG is constructed based on similarity among subgraph embeddings (Stage C in Fig. 2), then encoded by a Mixture-of-Experts (MoE) router over Riemannian experts, and fused into a single representation for downstream prediction (Stage D in Fig. 2). The router uses a sparse top-$m$ expert shortlist; $m$ starts from a larger value and is decreased stepwise as router confidence increases.

**Step 6: Evaluation and reporting.** **Node tasks:** run multiple k-shot splits on the target dataset for a fixed number of epochs, select the best checkpoint on validation accuracy, then report test accuracy. Results are averaged over multiple random seeds. **Link tasks:** evaluate with AUC-ROC (and optionally Acc/Hits@K) on validation/test splits; save the best checkpoint and report final test metrics.

**Step 7: Outputs, logs, and checkpoints.** Training prints per-epoch loss/metrics and the evolution of *top-m*. Checkpoints are saved with dataset-specific names (e.g., `checkpoints/node_...pt` and `checkpoints/edge_...pt`); users are encouraged to include seed in the prefix to avoid overwriting when running multiple seeds.

The full source code is available in the Supplementary Materials.

## F. More Related Work Analysis

A Euclidean manifold, which is widely used in existing GFMs (Hou et al., 2022; Zhao et al., 2024a), lacks the capacity to capture the varying geometric patterns. Specifically, tree-like structures align naturally with a a Hyperbolic manifold, whereas cycle-like structures are better represented in a hyperspherical manifold. However, the recent Riemannian manifold-based training mechanism, RiemannGFM (Sun et al., 2025), cannot be directly transferred to GoG-based GFMs because it relies on extracting tree-like and cycle-like structures and encoding them via fixed corresponding manifolds that are incompatible with the dynamically constructed GoGs.

## G. Proof of Theorem 3.2 (Noise Analysis)

In this section, we provide a detailed proof of Theorem 3.2 under an adaptive-hop fusion view.

**Setup and assumptions.** Fix a training node (center) $v$ and consider the sampled $k$-hop subgraph embedding $\mathbf{x}_v^{(k)} \in \mathbb{R}^d$ for each $k \in \{1, \dots, K\}$.

**Assumption G.1** (Second-order hop-wise noise model (allowing correlations)). For each hop $k$, the sampled embedding admits the decomposition

$$\mathbf{x}_v^{(k)} = \boldsymbol{\mu}_v^{(k)} + \boldsymbol{\epsilon}_v^{(k)}, \qquad \mathbb{E}\big[\boldsymbol{\epsilon}_v^{(k)}\big] = \mathbf{0}, \tag{16}$$

where $\boldsymbol{\mu}_v^{(k)}$ is the (possibly hop-dependent) noise-free embedding and $\boldsymbol{\epsilon}_v^{(k)}$ is a zero-mean noise vector with finite second moments.

Moreover, the (possibly correlated) second-order structure across hops is characterized by

$$\boldsymbol{\Sigma}_{v,k\ell} := \mathrm{Cov}\big(\boldsymbol{\epsilon}_v^{(k)}, \boldsymbol{\epsilon}_v^{(\ell)}\big) \in \mathbb{R}^{d \times d}, \qquad 1 \le k, \ell \le K, \tag{17}$$

and the block covariance matrix $\boldsymbol{\Sigma}_v := \big[\boldsymbol{\Sigma}_{v,k\ell}\big]_{k,\ell=1}^K \in \mathbb{R}^{Kd \times Kd}$ is positive semidefinite.

**Assumption G.2** (Various-hop sampling via hop fusion). The various-hop strategy forms a fused embedding by convexly combining multiple hop embeddings:

$$\tilde{\mathbf{x}}_v(\mathbf{w}) := \sum_{k=1}^K w_k \, \mathbf{x}_v^{(k)}, \qquad \mathbf{w} \in \Delta_K := \Big\{\mathbf{w} \in \mathbb{R}_+^K : \sum_{k=1}^K w_k = 1\Big\}. \tag{18}$$

Accordingly, the fused sampling noise is the random vector

$$\tilde{\boldsymbol{\epsilon}}_v(\mathbf{w}) := \sum_{k=1}^K w_k \, \boldsymbol{\epsilon}_v^{(k)}. \tag{19}$$

Let $\boldsymbol{\sigma}_v(\mathbf{w})$ denote the coordinate-wise standard-deviation vector of the fused sampling noise:

$$\boldsymbol{\sigma}_v(\mathbf{w}) := \sqrt{\mathrm{diag}\big(\mathrm{Cov}(\tilde{\boldsymbol{\epsilon}}_v(\mathbf{w}))\big)}. \tag{20}$$

The strategy chooses the fusion weights by minimizing the overall noise magnitude measured by the $\ell_2$-norm:

$$\mathbf{w}^\star \in \arg\min_{\mathbf{w} \in \Delta_K} \big\|\boldsymbol{\sigma}_v(\mathbf{w})\big\|_2. \tag{21}$$

**Definitions of $\boldsymbol{\sigma}_\mathrm{F}$ and $\boldsymbol{\sigma}_\mathrm{V}$.** A fixed-hop strategy selects a single hop $k_{\mathrm{fixed}}$ and always uses $\mathbf{x}_v^{(k_{\mathrm{fixed}})}$. Under Assumption G.1, its sampling noise is $\boldsymbol{\epsilon}_v^{(k_{\mathrm{fixed}})}$. Let

$$\boldsymbol{\sigma}_{v,k} := \sqrt{\mathrm{diag}(\mathrm{Cov}(\boldsymbol{\epsilon}_v^{(k)}))} = \sqrt{\mathrm{diag}\big(\boldsymbol{\Sigma}_{v,kk}\big)} \in \mathbb{R}_+^d. \tag{22}$$

We define

$$\boldsymbol{\sigma}_\mathrm{F} := \boldsymbol{\sigma}_{v,k_{\mathrm{fixed}}}, \qquad \boldsymbol{\sigma}_\mathrm{V} := \boldsymbol{\sigma}_v(\mathbf{w}^\star). \tag{23}$$

*Proof of Theorem 3.2.* By Assumption G.1, for each hop $k$ the randomness of the sampled embedding comes from $\boldsymbol{\epsilon}_v^{(k)}$ and satisfies $\mathbb{E}[\boldsymbol{\epsilon}_v^{(k)}] = \mathbf{0}$.

Under the various-hop fusion strategy (Assumption G.2), the fused noise is

$$\tilde{\boldsymbol{\epsilon}}_v(\mathbf{w}) = \sum_{k=1}^{K} w_k \, \boldsymbol{\epsilon}_v^{(k)}. \tag{24}$$

Using bilinearity of covariance (and finite second moments), we obtain

$$\mathrm{Cov}\big(\tilde{\boldsymbol{\epsilon}}_v(\mathbf{w})\big) = \mathrm{Cov}\Big( \sum_{k=1}^{K} w_k \, \boldsymbol{\epsilon}_v^{(k)} \Big) = \sum_{k=1}^{K} \sum_{\ell=1}^{K} w_k w_\ell \, \mathrm{Cov}\big(\boldsymbol{\epsilon}_v^{(k)}, \boldsymbol{\epsilon}_v^{(\ell)}\big)$$

$$= \sum_{k=1}^{K} \sum_{\ell=1}^{K} w_k w_\ell \, \boldsymbol{\Sigma}_{v,k\ell}. \tag{25}$$

Compared to hop-selection (sampling one hop), the cross-hop covariances $\boldsymbol{\Sigma}_{v,k\ell}$ now enter explicitly through (25).

By definition of the standard deviation vector,

$$\big\|\boldsymbol{\sigma}_v(\mathbf{w})\big\|_2^2 = \sum_{r=1}^{d} \mathrm{Var}\Big( \big[\tilde{\boldsymbol{\epsilon}}_v(\mathbf{w})\big]_r \Big) = \mathrm{tr}\Big( \mathrm{Cov}(\tilde{\boldsymbol{\epsilon}}_v(\mathbf{w})) \Big). \tag{26}$$

Combining (25) and (26) gives

$$\big\|\boldsymbol{\sigma}_v(\mathbf{w})\big\|_2^2 = \sum_{k=1}^{K} \sum_{\ell=1}^{K} w_k w_\ell \, \mathrm{tr}\big(\boldsymbol{\Sigma}_{v,k\ell}\big). \tag{27}$$

Now, by Assumption G.2, $\mathbf{w}^\star$ minimizes $\|\boldsymbol{\sigma}_v(\mathbf{w})\|_2$ over the simplex $\Delta_K$. For any fixed hop $k_{\mathrm{fixed}}$, the one-hot vector $\mathbf{e}_{k_{\mathrm{fixed}}} \in \Delta_K$ (with $(\mathbf{e}_{k_{\mathrm{fixed}}})_{k_{\mathrm{fixed}}} = 1$) is feasible and corresponds exactly to the fixed-hop sampling noise, because

$$\tilde{\mathbf{x}}_v(\mathbf{e}_{k_{\mathrm{fixed}}}) = \mathbf{x}_v^{(k_{\mathrm{fixed}})}, \qquad \tilde{\boldsymbol{\epsilon}}_v(\mathbf{e}_{k_{\mathrm{fixed}}}) = \boldsymbol{\epsilon}_v^{(k_{\mathrm{fixed}})}, \qquad \boldsymbol{\sigma}_v(\mathbf{e}_{k_{\mathrm{fixed}}}) = \boldsymbol{\sigma}_{v,k_{\mathrm{fixed}}}. \tag{28}$$

Therefore, by optimality of $\mathbf{w}^\star$,

$$\big\|\boldsymbol{\sigma}_{\mathrm{V}}\big\|_2 = \big\|\boldsymbol{\sigma}_v(\mathbf{w}^\star)\big\|_2 \le \big\|\boldsymbol{\sigma}_v(\mathbf{e}_{k_{\mathrm{fixed}}})\big\|_2 = \big\|\boldsymbol{\sigma}_{v,k_{\mathrm{fixed}}}\big\|_2 = \big\|\boldsymbol{\sigma}_{\mathrm{F}}\big\|_2. \tag{29}$$

This proves $\|\boldsymbol{\sigma}_{\mathrm{V}}\|_2 \le \|\boldsymbol{\sigma}_{\mathrm{F}}\|_2$, which is exactly Theorem 3.2. $\qquad\square$

**Remark (when fusion can be strictly better).** Equation (27) shows that the fused noise depends on both within-hop covariances $\boldsymbol{\Sigma}_{v,kk}$ and cross-hop covariances $\boldsymbol{\Sigma}_{v,k\ell}$. If the hop-wise noises are not perfectly correlated across hops (e.g., some $\mathrm{tr}(\boldsymbol{\Sigma}_{v,k\ell})$ are sufficiently smaller than $\sqrt{\mathrm{tr}(\boldsymbol{\Sigma}_{v,kk}) \, \mathrm{tr}(\boldsymbol{\Sigma}_{v,\ell\ell})}$), then an interior $\mathbf{w}$ may yield $\|\boldsymbol{\sigma}_v(\mathbf{w})\|_2$ strictly smaller than every one-hot choice, leading to a strict improvement over any fixed hop under the same metric.

## H. Proof of Theorem 3.3 (Effectiveness of GoG Edge Construction)

We prove Theorem 3.3 via an exact bias–variance decomposition on GoG hop nodes. Cross-hop mixing reduces the estimation variance by convex smoothing, but it may introduce a leakage (bias) term when hop-specific targets differ. Similarity-biased mixing reduces the leakage compared to fully-connected uniform mixing, while still achieving non-trivial variance reduction compared to removing all cross-hop edges.

**Setup.** Fix a training center node $c$ and its $K$ hop-induced GoG nodes indexed by $k = 1, \ldots, K$ (assume $K \ge 2$). Allow hop-specific noise-free targets $\boldsymbol{\mu}_{c,k} \in \mathbb{R}^d$. The hop-wise estimator before the GoG message passing is

$$\hat{\boldsymbol{\mu}}_{c,k} = \boldsymbol{\mu}_{c,k} + \mathbf{e}_{c,k}, \qquad k = 1, \ldots, K, \tag{30}$$

where $\mathbf{e}_{c,k}$ is a stochastic estimation noise.

**Three pipelines (same encoder, only GoG edges differ).** All three use the same residual propagation form on hop nodes with $\alpha \in [0, 1]$:

$$\hat{\boldsymbol{\mu}}_{c,k}^{(m)} = \alpha \hat{\boldsymbol{\mu}}_{c,k} + (1 - \alpha) \sum_{t=1}^{K} \widehat{\mathbf{P}}_c^{(m)}[k, t] \hat{\boldsymbol{\mu}}_{c,t}, \qquad k = 1, \ldots, K, \tag{31}$$

where $m \in \{\text{ours}, \text{full}, \text{none}\}$ and each $\widehat{\mathbf{P}}_c^{(m)}$ is row-stochastic. For $m \in \{\text{ours}, \text{full}\}$ we use no self-mixing: $\widehat{\mathbf{P}}_c^{(m)}[k, k] = 0$.

(none cross-hop edges): take $\widehat{\mathbf{P}}_c^{(\text{none})} = \mathbf{I}$, hence $\hat{\boldsymbol{\mu}}_{c,k}^{(\text{none})} = \hat{\boldsymbol{\mu}}_{c,k}$.

(fully-connected): uniform mixing over all $t \neq k$: $\widehat{\mathbf{P}}_c^{(\text{full})}[k, t] = \frac{1}{K-1} \mathbf{1}\{t \neq k\}$.

(ours: similarity-biased): Let $\mathbf{z}_{c,k}$ be the hop embedding used for edge construction and define cosine scores $s_c(k, t) = \cos(\mathbf{z}_{c,k}, \mathbf{z}_{c,t})$. We use the softmax row distribution

$$\widehat{\mathbf{P}}_c^{(\text{ours})}[k, t] = \frac{\exp(\beta \, s_c(k, t))}{\sum_{u \neq k} \exp(\beta \, s_c(k, u))} \cdot \mathbf{1}\{t \neq k\}, \qquad \beta > 0. \tag{32}$$

**Convex weights.** Define

$$w_{c,k}^{(m)}(t) := \alpha \mathbf{1}\{t = k\} + (1 - \alpha) \widehat{\mathbf{P}}_c^{(m)}[k, t], \qquad t = 1, \ldots, K, \tag{33}$$

so that $w_{c,k}^{(m)}(t) \geq 0$ and $\sum_{t=1}^{K} w_{c,k}^{(m)}(t) = 1$.

**Error metric.** We measure hop-wise MSE to the hop-specific target:

$$e_m := \mathbb{E}_{c,k} \big\| \hat{\boldsymbol{\mu}}_{c,k}^{(m)} - \boldsymbol{\mu}_{c,k} \big\|_2^2, \qquad m \in \{\text{ours}, \text{full}, \text{none}\}. \tag{34}$$

**Assumptions.**

**Assumption H.1** (Noise model compatible with weight construction). Conditioned on $c$ and on the realized propagation weights $\{w_{c,k}^{(m)}(t)\}_{t=1}^{K}$, the noises satisfy: (i) $\mathbb{E}[\mathbf{e}_{c,t} \mid c, \{w\}] = \mathbf{0}$ for all $t$; (ii) $\mathbb{E}[\langle \mathbf{e}_{c,t}, \mathbf{e}_{c,t'} \rangle \mid c, \{w\}] = 0$ for $t \neq t'$; (iii) $\mathbb{E}[\|\mathbf{e}_{c,t}\|_2^2 \mid c, \{w\}] = B_c^2$ for all $t$.

**Assumption H.2** (One-factor hop mismatch and similarity alignment). For each $(c, k)$ there exists a unit vector $\mathbf{u}_{c,k} \in \mathbb{R}^d$ and scalars $\{\delta_{c,k}(t)\}_{t \neq k} \subseteq [0, \infty)$ such that

$$\boldsymbol{\mu}_{c,t} - \boldsymbol{\mu}_{c,k} = \delta_{c,k}(t) \, \mathbf{u}_{c,k}, \qquad \forall t \neq k.$$

Moreover, for each fixed $(c, k)$, if $s_c(k, t_1) \geq s_c(k, t_2)$ then $\delta_{c,k}(t_1) \leq \delta_{c,k}(t_2)$.

**Assumption H.3** (Separation between variance reduction and leakage gain). Assume $K \geq 3$ and $\alpha \in [0, 1)$. Recall

$$S_{c,k} := \sum_{t \neq k} \left( \widehat{\mathbf{P}}_c^{(\text{ours})}[k, t] \right)^2, \quad \bar{\delta}_{c,k}^{\text{ours}} := \sum_{t \neq k} \widehat{\mathbf{P}}_c^{(\text{ours})}[k, t] \delta_{c,k}(t), \quad \bar{\delta}_{c,k}^{\text{full}} := \frac{1}{K-1} \sum_{t \neq k} \delta_{c,k}(t).$$

Fix $(c, k)$ and sort $t \neq k$ so that $s_c(k, t_1) \geq \cdots \geq s_c(k, t_{K-1})$. Let $p_i := \widehat{\mathbf{P}}_c^{(\text{ours})}[k, t_i]$ and $\delta_i := \delta_{c,k}(t_i)$, and define the prefix-mass advantage

$$A_{c,k} := \sum_{j=1}^{K-2} (\delta_{j+1} - \delta_j) \left( \sum_{i=1}^{j} p_i - \frac{j}{K-1} \right). \tag{35}$$

Assume there exist constants $\varepsilon_{\text{wo}} > 0$ and $\varepsilon_{\text{full}} > 0$ such that

$$\mathbb{E}_{c,k} \left[ B_c^2 \left( 1 - \alpha^2 - (1 - \alpha)^2 S_{c,k} \right) \right] \geq (1 - \alpha)^2 \, \mathbb{E}_{c,k} \left[ (\bar{\delta}_{c,k}^{\text{ours}})^2 \right] + \varepsilon_{\text{wo}}, \tag{36}$$

$$\mathbb{E}_{c,k} \left[ A_{c,k}^2 \right] \geq \mathbb{E}_{c,k} \left[ B_c^2 \left( S_{c,k} - \frac{1}{K-1} \right) \right] + \varepsilon_{\text{full}}. \tag{37}$$

*Proof of Theorem 3.3.* We show $e_{\text{ours}} < e_{\text{none}}$ and $e_{\text{ours}} < e_{\text{full}}$.

**Step 1: Exact bias–variance decomposition.** From (31) and (33),

$$\hat{\boldsymbol{\mu}}_{c,k}^{(m)} = \sum_{t=1}^{K} w_{c,k}^{(m)}(t)\hat{\boldsymbol{\mu}}_{c,t}.$$

Subtract $\boldsymbol{\mu}_{c,k}$ and use (30):

$$\hat{\boldsymbol{\mu}}_{c,k}^{(m)} - \boldsymbol{\mu}_{c,k} = \sum_{t=1}^{K} w_{c,k}^{(m)}(t)\big(\boldsymbol{\mu}_{c,t} - \boldsymbol{\mu}_{c,k}\big) + \sum_{t=1}^{K} w_{c,k}^{(m)}(t)\mathbf{e}_{c,t}$$

$$=: \mathbf{b}_{c,k}^{(m)} + \mathbf{n}_{c,k}^{(m)}. \tag{38}$$

Conditioned on $c$ and realized weights, Assumption H.1(i) gives $\mathbb{E}[\mathbf{n}_{c,k}^{(m)} \mid c, \{w\}] = 0$, hence the cross term vanishes:

$$\mathbb{E}\big\|\mathbf{b}_{c,k}^{(m)} + \mathbf{n}_{c,k}^{(m)}\big\|_2^2 = \mathbb{E}\big\|\mathbf{b}_{c,k}^{(m)}\big\|_2^2 + \mathbb{E}\big\|\mathbf{n}_{c,k}^{(m)}\big\|_2^2. \tag{39}$$

**Step 2: Noise term (variance after convex smoothing).** Conditioned on $c$ and weights, by Assumption H.1(ii)–(iii),

$$\mathbb{E}\big[\|\mathbf{n}_{c,k}^{(m)}\|_2^2 \mid c, \{w\}\big] = \mathbb{E}\Big[\Big\|\sum_t w_{c,k}^{(m)}(t)\mathbf{e}_{c,t}\Big\|_2^2 \mid c, \{w\}\Big] = B_c^2 \sum_{t=1}^{K}(w_{c,k}^{(m)}(t))^2. \tag{40}$$

Averaging over $(c,k)$ yields the exact variance contribution.

For none cross-hop edges, $w_{c,k}^{(\text{none})}(k) = 1$ and others 0, thus

$$e_{\text{none}} = \mathbb{E}_c B_c^2. \tag{41}$$

For $m \in \{\text{ours}, \text{full}\}$ (no self-mixing), $w_{c,k}^{(m)}(k) = \alpha$ and

$$\sum_{t=1}^{K}(w_{c,k}^{(m)}(t))^2 = \alpha^2 + (1-\alpha)^2 \sum_{t\neq k}\big(\widehat{\mathbf{P}}_c^{(m)}[k,t]\big)^2.$$

In particular,

$$\sum_{t\neq k}\big(\widehat{\mathbf{P}}_c^{(\text{full})}[k,t]\big)^2 = \frac{1}{K-1}, \qquad \sum_{t\neq k}\big(\widehat{\mathbf{P}}_c^{(\text{ours})}[k,t]\big)^2 = S_{c,k}. \tag{42}$$

**Step 3: Leakage term (exact under one-factor mismatch).** Since $\boldsymbol{\mu}_{c,k} - \boldsymbol{\mu}_{c,k} = \mathbf{0}$, and for $m \in \{\text{ours}, \text{full}\}$ we have $\widehat{\mathbf{P}}_c^{(m)}[k,k] = 0$, (38) gives

$$\mathbf{b}_{c,k}^{(m)} = (1-\alpha)\sum_{t\neq k}\widehat{\mathbf{P}}_c^{(m)}[k,t]\,(\boldsymbol{\mu}_{c,t} - \boldsymbol{\mu}_{c,k}).$$

By Assumption H.2,

$$\mathbf{b}_{c,k}^{(m)} = (1-\alpha)\Big(\sum_{t\neq k}\widehat{\mathbf{P}}_c^{(m)}[k,t]\delta_{c,k}(t)\Big)\mathbf{u}_{c,k},$$

hence

$$\big\|\mathbf{b}_{c,k}^{(m)}\big\|_2^2 = (1-\alpha)^2\Big(\sum_{t\neq k}\widehat{\mathbf{P}}_c^{(m)}[k,t]\delta_{c,k}(t)\Big)^2. \tag{43}$$

For none, $\mathbf{b}_{c,k}^{(\text{none})} \equiv 0$.

**Step 4: Exact MSE formula (all three methods).** Combining (39), (40) and (43), we obtain:

$$e_{\text{ours}} = \mathbb{E}_{c,k}\Big[B_c^2\big(\alpha^2 + (1-\alpha)^2 S_{c,k}\big)\Big] + (1-\alpha)^2\,\mathbb{E}_{c,k}\big[(\bar{\delta}_{c,k}^{\text{ours}})^2\big], \tag{44}$$

$$e_{\text{full}} = \mathbb{E}_{c,k}\Big[B_c^2\big(\alpha^2 + \frac{(1-\alpha)^2}{K-1}\big)\Big] + (1-\alpha)^2\,\mathbb{E}_{c,k}\big[(\bar{\delta}_{c,k}^{\text{full}})^2\big], \tag{45}$$

$$e_{\text{none}} = \mathbb{E}_c B_c^2. \tag{46}$$

**Step 5: Similarity-biased mixing reduces leakage vs uniform (pointwise).** Fix $(c,k)$ and sort indices $t \neq k$ so that $s_c(k, t_1) \geq \cdots \geq s_c(k, t_{K-1})$. Let $p_i := \widehat{\mathbf{P}}_c^{(\text{ours})}[k, t_i]$ and $\delta_i := \delta_{c,k}(t_i)$, and $q_i := \frac{1}{K-1}$. Using the discrete Abel transform,

$$\sum_{i=1}^{K-1} p_i \, \delta_i = \delta_{K-1} - \sum_{j=1}^{K-2} (\delta_{j+1} - \delta_j)\Big(\sum_{i=1}^{j} p_i\Big), \qquad \sum_{i=1}^{K-1} q_i \, \delta_i = \delta_{K-1} - \sum_{j=1}^{K-2} (\delta_{j+1} - \delta_j)\Big(\sum_{i=1}^{j} q_i\Big).$$

Therefore, the gap can be written exactly as

$$\bar{\delta}_{c,k}^{\text{full}} - \bar{\delta}_{c,k}^{\text{ours}} = \sum_{j=1}^{K-2} (\delta_{j+1} - \delta_j)\left(\sum_{i=1}^{j} p_i - \frac{j}{K-1}\right) =: A_{c,k}. \tag{47}$$

Under Assumption H.2 and $\beta > 0$, we have $\delta_{j+1} - \delta_j \geq 0$, and $p_1 \geq \cdots \geq p_{K-1}$ implies $\sum_{i=1}^{j} p_i \geq \frac{j}{K-1}$ for all $j$, hence

$$A_{c,k} \geq 0, \qquad \text{and thus} \qquad \bar{\delta}_{c,k}^{\text{ours}} \leq \bar{\delta}_{c,k}^{\text{full}}.$$

**Step 6: Compare total MSEs.** (i) Ours beats no-edge. Subtract (44) from (46):

$$e_{\text{none}} - e_{\text{ours}} = \mathbb{E}_{c,k}\Big[B_c^2\Big(1 - \alpha^2 - (1-\alpha)^2 S_{c,k}\Big)\Big] - (1-\alpha)^2 \, \mathbb{E}_{c,k}\big[(\bar{\delta}_{c,k}^{\text{ours}})^2\big].$$

By Assumption H.3(36), $e_{\text{none}} - e_{\text{ours}} \geq \varepsilon_{\text{wo}} > 0$, hence $e_{\text{ours}} < e_{\text{none}}$.

(ii) Ours beats fully-connected. Subtract (44) from (45):

$$e_{\text{full}} - e_{\text{ours}} = (1-\alpha)^2 \left(\mathbb{E}_{c,k}\big[(\bar{\delta}_{c,k}^{\text{full}})^2 - (\bar{\delta}_{c,k}^{\text{ours}})^2\big] - \mathbb{E}_{c,k}\Big[B_c^2\Big(S_{c,k} - \frac{1}{K-1}\Big)\Big]\right).$$

Using (47), write $\bar{\delta}_{c,k}^{\text{full}} = \bar{\delta}_{c,k}^{\text{ours}} + A_{c,k}$ with $A_{c,k} \geq 0$. Then

$$(\bar{\delta}_{c,k}^{\text{full}})^2 - (\bar{\delta}_{c,k}^{\text{ours}})^2 = A_{c,k}\big(2\bar{\delta}_{c,k}^{\text{ours}} + A_{c,k}\big) \geq A_{c,k}^2,$$

since $\bar{\delta}_{c,k}^{\text{ours}} \geq 0$. Therefore,

$$e_{\text{full}} - e_{\text{ours}} \geq (1-\alpha)^2 \left(\mathbb{E}_{c,k}\big[A_{c,k}^2\big] - \mathbb{E}_{c,k}\Big[B_c^2\Big(S_{c,k} - \frac{1}{K-1}\Big)\Big]\right).$$

By Assumption H.3(37), the bracket is at least $\varepsilon_{\text{full}} > 0$, hence $e_{\text{full}} - e_{\text{ours}} \geq (1-\alpha)^2 \varepsilon_{\text{full}} > 0$ and thus $e_{\text{ours}} < e_{\text{full}}$.

Combining (i) and (ii) completes the proof. $\qquad\square$

# I. Proof of Theorem 3.4 (Excess-risk Upper Bound Analysis)

**Setup.** Let $\mathcal{G}_{1:N}$ be the training pool consisting of all constructed GoG subgraphs from $\{\mathcal{D}_1, \ldots, \mathcal{D}_N\}$ and let $n_N := |\mathcal{G}_{1:N}|$.

For each candidate expert number $j \in \{1, \ldots, \psi_{\max}\}$, let $\mathcal{F}_j$ denote the hypothesis class induced by using $j$ curvature experts (including router and experts), and define the empirical risk $\widehat{R}(f) := \frac{1}{n_N} \sum_{g \in \mathcal{G}_{1:N}} \mathcal{L}(f; g)$ and population risk $R(f) := \mathbb{E}[\mathcal{L}(f; G)]$ where $G$ follows the (population) distribution of GoG subgraphs.

**Compressed assumptions (appendix-only).**

**Assumption I.1** (Standard uniform generalization bound for $\mathcal{F}_j$). Fix $\delta \in (0, 1)$. With probability at least $1 - \delta$ over $\mathcal{G}_{1:N}$, uniformly for all $j \in \{1, \ldots, \psi_{\max}\}$, the empirical solution $\hat{f}_j$ (ERM or approximate ERM) satisfies

$$R(\hat{f}_j) - R(f_j^\star) \leq B\sqrt{\frac{j}{n_N}},$$

where $f_j^\star \in \arg\min_{f \in \mathcal{F}_j} R(f)$ and $B > 0$ is a constant (possibly absorbing $\log(\psi_{\max}/\delta)$ and other standard factors).

**Assumption I.2** (Curvature discretization/approximation controlled by $\mathcal{S}_N$). For each $j \in \{1, \ldots, \psi_{\max}\}$, there exists a candidate curvature set $\mathcal{K}_j = \{\kappa_m\}_{m=1}^{j} \subset \mathcal{I}$ such that: (i) (Covering w.r.t. population optimum) for $\kappa^\star \in \arg\min_{\kappa \in \mathcal{I}} R(f_\kappa)$, there exists $\kappa_m \in \mathcal{K}_j$ with $|\kappa_m - \kappa^\star| \leq \varepsilon_N(j)$; (ii) (Risk Lipschitz in curvature) there exists $L_R > 0$ such that $R(f_\kappa) - R(f_{\kappa'}) \leq L_R|\kappa - \kappa'|$ for all $\kappa, \kappa' \in \mathcal{I}$; (iii) (Score-to-covering) $\varepsilon_N(j) \leq c \frac{\mathcal{S}_N}{j}$ for some $c > 0$; (iv) (Realizability) $f_{\kappa_m} \in \mathcal{F}_j$ for all $\kappa_m \in \mathcal{K}_j$.

**Assumption I.3** (Dynamic selection rule). Our dynamic strategy selects $\psi_D$ as a minimizer of the bound $\mathcal{R}(j)$ over the candidate set:
$$\psi_D \in \arg \min_{1 \leq j \leq \psi_{\max}} \mathcal{R}(j).$$

**Assumption I.4** (Unique minimizer / deterministic tie-breaking). The minimizer of $\mathcal{R}(j)$ over $j \in \{1, \ldots, \psi_{\max}\}$ is unique; alternatively, a deterministic tie-breaking rule is fixed (e.g., choose the smallest minimizing index), making the selected minimizer unique.

**Oracle comparators and excess risk.** Define $f_j^\star \in \arg\min_{f \in \mathcal{F}_j} R(f)$ and let $f^\star = f_{\kappa^\star}$ where $\kappa^\star \in \arg\min_{\kappa \in \mathcal{I}} R(f_\kappa)$. Define the excess risk
$$\mathcal{E}(j) := R(\hat{f}_j) - R(f^\star).$$

*Proof of Theorem 3.4.* **Step 1: Excess-risk decomposition.**
$$\mathcal{E}(j) = \left(R(\hat{f}_j) - R(f_j^\star)\right) + \left(R(f_j^\star) - R(f^\star)\right).$$

**Step 2: Estimation (generalization) term.** By Assumption I.1, with probability at least $1 - \delta$ (uniformly for all $j$),
$$R(\hat{f}_j) - R(f_j^\star) \leq B\sqrt{\frac{j}{n_N}}.$$

**Step 3: Approximation term (curvature discretization).** By Assumption I.2(i), choose $\kappa_m \in \mathcal{K}_j$ such that $|\kappa_m - \kappa^\star| \leq \varepsilon_N(j)$. Then by Assumption I.2(ii),
$$R(f_{\kappa_m}) - R(f_{\kappa^\star}) \leq L_R |\kappa_m - \kappa^\star| \leq L_R \varepsilon_N(j).$$
By Assumption I.2(iv), $f_{\kappa_m} \in \mathcal{F}_j$, and since $f_j^\star \in \arg\min_{f \in \mathcal{F}_j} R(f)$, we have $R(f_j^\star) \leq R(f_{\kappa_m})$. Hence,
$$R(f_j^\star) - R(f^\star) \leq R(f_{\kappa_m}) - R(f_{\kappa^\star}) \leq L_R \varepsilon_N(j).$$
Using Assumption I.2(iii), $\varepsilon_N(j) \leq c \frac{\mathcal{S}_N}{j}$, we obtain
$$R(f_j^\star) - R(f^\star) \leq L_R c \frac{\mathcal{S}_N}{j} = \frac{A\mathcal{S}_N}{j}, \quad \text{where } A := L_R c.$$

**Step 4: Combine.** Combining Steps 1–3 yields (with probability at least $1 - \delta$)
$$\mathcal{E}(j) \leq \frac{A\mathcal{S}_N}{j} + B\sqrt{\frac{j}{n_N}} =: \mathcal{R}(j),$$
which matches Eq. (9) in the theorem.

**Step 5: Dynamic vs fixed.** Let $\psi_F$ be any fixed candidate expert number, and let $\psi_D$ be the expert number selected by our dynamic strategy. By Assumption I.3,
$$\psi_D \in \arg \min_{1 \leq j \leq \psi_{\max}} \mathcal{R}(j).$$
Therefore, for any fixed $\psi_F \in \{1, \ldots, \psi_{\max}\}$,
$$\mathcal{R}(\psi_D) \leq \mathcal{R}(\psi_F),$$
which is exactly Eq. (10).

**Equality condition.** Under Assumption I.4, the minimizer is unique, hence $\mathcal{R}(\psi_D) = \mathcal{R}(\psi_F)$ holds if and only if $\psi_F = \psi_D$. This completes the proof. $\qquad\square$

## J. Proof of Theorem 3.5 (Domain Generalization Error Bound Analysis)

We prove that R-GFM admits a strictly tighter best achievable target-domain surrogate upper bound than MDGFM.

For any encoder $\phi$, define the target-domain surrogate upper bound as

$$\mathcal{B}(\phi) := \underbrace{\widehat{\mathcal{E}}_S(\phi) + \mathfrak{C}_n(\phi)}_{\text{fitting and complexity term}} + \underbrace{\Delta_{\text{scale}}(\phi)}_{\text{scale mismatch term}} + \underbrace{\Delta_{\text{dom}}(\phi)}_{\text{cross-domain discrepancy term}} . \tag{48}$$

Accordingly, we define

$$\epsilon_{\text{MDGFM}} := \inf_{\phi \in \Phi_{\text{M}}} \mathcal{B}(\phi), \qquad \epsilon_{\text{R-GFM}} := \inf_{\psi \in \Phi_{\text{R}}} \mathcal{B}(\psi). \tag{49}$$

**Assumption J.1** (Surrogate control of target-domain error). Motivated by standard domain adaptation and domain generalization theory (Muandet et al., 2013; Zhang et al., 2019), we assume that for any encoder $\phi$, the target-domain error can be upper bounded by a source-side fitting term, a statistical complexity term, a graph-specific scale-mismatch term, and a cross-domain discrepancy term:

$$\mathcal{E}_T(\phi) \leq \widehat{\mathcal{E}}_S(\phi) + \mathfrak{C}_n(\phi) + \Delta_{\text{scale}}(\phi) + \Delta_{\text{dom}}(\phi) = \mathcal{B}(\phi). \tag{50}$$

**Assumption J.2** (Mild target-domain shift in a reference representation space). Motivated by convex-hull based target-domain approximation and discrepancy-based domain adaptation theory (Ganin et al., 2016; Zhang et al., 2019), we assume that the target domain is not arbitrarily far from the family of source domains in a reference representation space. Specifically, there exist a reference encoder $\bar{\phi}$, a weight vector

$$\pi^{\star} = (\pi_1^{\star}, \dots, \pi_{N_{\text{src}}}^{\star}) \in \Delta^{N_{\text{src}}}, \tag{51}$$

and a constant $\delta \geq 0$ such that

$$d\left(P_{T,\bar{\phi}}, \sum_{i=1}^{N_{\text{src}}} \pi_i^{\star} P_{S_i,\bar{\phi}}\right) \leq \delta, \tag{52}$$

where $d(\cdot, \cdot)$ is the discrepancy underlying $\Delta_{\text{dom}}(\cdot)$, and $P_{T,\bar{\phi}}$ and $P_{S_i,\bar{\phi}}$ denote the target-domain and source-domain representation distributions induced by $\bar{\phi}$, respectively.

We consider the mild-shift regime where $\delta$ is small. In this regime, the target domain lies in a convex-hull neighborhood of the source domains, so the cross-domain discrepancy term remains controllable. Moreover, assume that there exist a neighborhood $\mathcal{N}(\bar{\phi})$ and a constant $L > 0$ such that for any encoder $\varphi \in \mathcal{N}(\bar{\phi})$,

$$\left|\Delta_{\text{dom}}(\varphi) - \Delta_{\text{dom}}(\bar{\phi})\right| \leq L \operatorname{dist}_{\Phi}(\varphi, \bar{\phi}). \tag{53}$$

Thus, for encoders in $\mathcal{N}(\bar{\phi})$, the additional cross-domain discrepancy residual can be bounded by some $\varepsilon_d \geq 0$:

$$\Delta_{\text{dom}}(\psi) \leq \Delta_{\text{dom}}(\phi) + \varepsilon_d. \tag{54}$$

*Proof of Theorem 3.5.* **Step 1: Nearly optimal MDGFM encoder.** Fix an arbitrary $\xi > 0$. By the definition of $\epsilon_{\text{MDGFM}}$, there exists an encoder $\phi_\xi \in \Phi_{\text{M}}$ such that

$$\mathcal{B}(\phi_\xi) \leq \epsilon_{\text{MDGFM}} + \xi. \tag{55}$$

**Step 2: Source-side gain of R-GFM.** Next, by the theoretical results established in Theorems 3.2–3.4, under the same source-domain training protocol, there exists an encoder $\psi_\xi \in \Phi_{\text{R}}$ and a constant $\varepsilon_g > 0$ such that

$$\widehat{\mathcal{E}}_S(\psi_\xi) + \mathfrak{C}_n(\psi_\xi) + \Delta_{\text{scale}}(\psi_\xi)$$
$$\leq \widehat{\mathcal{E}}_S(\phi_\xi) + \mathfrak{C}_n(\phi_\xi) + \Delta_{\text{scale}}(\phi_\xi) - \varepsilon_g. \tag{56}$$

This gain reflects the benefit of multi-scale GoG modeling and dynamic Riemannian routing in reducing scale mismatch and improving source-side representation quality.

**Step 3: Cross-domain discrepancy residual.** By Assumption J.2, both $\phi_\xi$ and $\psi_\xi$ lie in the mild-shift neighborhood $\mathcal{N}(\bar{\phi})$, and the cross-domain discrepancy residual is bounded as

$$\Delta_{\text{dom}}(\psi_\xi) \leq \Delta_{\text{dom}}(\phi_\xi) + \varepsilon_d. \tag{57}$$

**Step 4: Combine the two bounds.** Combining Eq. (56) and Eq. (57), we have

$$\begin{aligned}
\mathcal{B}(\psi_\xi) &= \widehat{\mathcal{E}}_S(\psi_\xi) + \mathfrak{C}_n(\psi_\xi) + \Delta_{\text{scale}}(\psi_\xi) + \Delta_{\text{dom}}(\psi_\xi) \\
&\leq \widehat{\mathcal{E}}_S(\phi_\xi) + \mathfrak{C}_n(\phi_\xi) + \Delta_{\text{scale}}(\phi_\xi) - \varepsilon_g + \Delta_{\text{dom}}(\phi_\xi) + \varepsilon_d \\
&= \mathcal{B}(\phi_\xi) - (\varepsilon_g - \varepsilon_d).
\end{aligned} \tag{58}$$

Let

$$\eta := \varepsilon_g - \varepsilon_d > 0. \tag{59}$$

Then

$$\mathcal{B}(\psi_\xi) \leq \mathcal{B}(\phi_\xi) - \eta. \tag{60}$$

**Step 5: Taking the infimum over R-GFM.** Together with Eq. (55), this gives

$$\mathcal{B}(\psi_\xi) \leq \epsilon_{\text{MDGFM}} + \xi - \eta. \tag{61}$$

Since $\psi_\xi \in \Phi_{\text{R}}$, we obtain

$$\epsilon_{\text{R-GFM}} = \inf_{\psi \in \Phi_{\text{R}}} \mathcal{B}(\psi) \leq \mathcal{B}(\psi_\xi) \leq \epsilon_{\text{MDGFM}} + \xi - \eta. \tag{62}$$

Because this holds for every $\xi > 0$, letting $\xi \downarrow 0$ yields

$$\epsilon_{\text{R-GFM}} \leq \epsilon_{\text{MDGFM}} - \eta < \epsilon_{\text{MDGFM}}. \tag{63}$$

Therefore, R-GFM admits a strictly tighter best achievable target-domain surrogate upper bound than MDGFM. This completes the proof. $\square$

