# OpenReview forum: "Learning Graph Foundation Models on Riemannian Graph-of-Graphs"
_ICML.cc/2026/Conference — ICML 2026 regular_

### Official Review · Reviewer_uePc · 2026-03-04

**Soundness:** 3
**Presentation:** 3
**Significance:** 3
**Originality:** 3
**Overall Recommendation:** 4
**Confidence:** 3

**Summary:**

This paper proposes a novel method to build Graph Foundation Models (GFMs) that doesn't rely on fixed-hop subgraph sampling. The authors argue that current GFMs fail because they use a fixed receptive field which doesn't work well for different graphs that need different scales of context. Instead, they propose a "Graph-of-Graphs" (GOG) framework. They map different graph samplings (from local to global) onto a Riemannian manifold. By doing this, they can treat the structural scale as a learnable parameter. The model uses a manifold-based routing process to learn how to move between these structural scales and aggregate for downstream prediction.

**Compliance With Llm Reviewing Policy:**

Affirmed.

**Final Justification:**

The rebuttal reinforced my prior assessment that this is a solid paper with an interesting idea to tackle fixed-hop subgraph sampling in GFM subfield. I keep my score at 4.

**Key Questions For Authors:**

1) Can you provide the training time and inference latency for R-GFM compared to standard fixed-hop GNNs and GFMs? Does the Riemannian optimization significantly slow down the pretraining process on massive datasets?

2) How does this method perform on graph-level prediction (e.g. molecular graph property prediction)? Are there any benefits of pretraining on large sets of molecules on downstream molecular property prediction?

3) How does this method comapre to other mixed-hops aggregation method (e.g. MixHop, Abu-El-Haija et al. 2019)

**Limitations:**

Yes

**Strengths And Weaknesses:**

**Strengths**:
- The paper addresses a real problem where "one-size-fits-all" subgraph sampling limits how foundation models work on different types of graphs.

- The proposed idea seems novel and original.

- Extensive experiments on datasets across different domains.

- The authors report strong performance on all datasets, outperforming baselines.

- Ablation studies demonstrate the importance of core parts.

**Weaknesses**:

- The main weakness the reviewer sees is the added computational complexity and potential training instabilities. The authors don't discuss the added complexity.

- The authors only include evaluations for node classification and link prediction. Graph-level classification is missing.

- Related work section seems a bit limited. What about other methods with dynamic hops (e.g. MixHop, Abu-El-Haija et al. 2019)?

---

> ### Author Rebuttal · Authors · 2026-03-30
>
> # Response to W1&Q1 (Efficiency: Training Time / Inference Latency)
> Thank you for your advice. Following your suggestion, we report the training time and inference latency in the table below. The training datasets were uniformly set to Wisconsin, Cornell, Citeseer, Pubmed, Computers, Photos, and Texas, while the inference latency was evaluated on Cora.
>
> |Model|Training time (s)|Inference time (ms)|
> |-|-|-|
> |MDGFM|142.77|90.42|
> |SAMGPT|243.31|56.16|
> |GCOPE|3162.30|840.62|
> |RiemannGFM|2740.83|882.74|
> |R-GFM|1187.11|763.32|
>
> Based on these results, **MDGFM** and **SAMGPT** are indeed faster in training/inference. However, they both rely on a full-graph training pipeline and become **out-of-memory** on larger graphs as shown in **Table 2** of the manuscript. In contrast, **R-GFM** remains trainable on larger graphs through subgraph sampling and GoG construction. Compared with **RiemannGFM**, R-GFM is substantially more efficient in both training and inference. Moreover, although its efficiency is on the same order of magnitude as **GCOPE**, R-GFM achieves higher accuracy, indicating a more favorable trade-off between effectiveness and computational cost.
>
> Additionally, our model converges stably. As shown in the table below, as the number of epochs increases, the losses of the pretraining and tuning all converge to relatively small values.
>
> |Epoch|Pretraining Loss|
> |-|-|
> |1|4.0361|
> |10|2.5341|
> |20|2.4084|
> |50|2.2714|
> |100|2.1523|
>
> |Epoch|Tuning Loss|
> |-|-|
> |1|1.7332|
> |10|0.1689|
> |20|0.0021|
> |60|0.0002|
> |100|0.0002|
>
> For the Riemannian optimization, we acknowledge that it introduces additional routing overhead. However, this overhead is **marginal** relative to the total training time in practice, as shown in the table below.
> |Forward time per step (ms)|Routing time per step (ms)| Routing Time / Total Training Time|
> |-|-|-|
> |66.32|3.35|5.05%|
>
> We will add these results to the revised manuscript, better illustrating the favorable effectiveness-efficiency trade-off of our model.
>
> # Response to W2&Q2 (Graph-level Prediction / Molecular Property Prediction)
> Following your suggestion, we additionally conduct a transfer pretraining experiment on molecular graphs to examine whether R-GFM is also effective in this setting.
>
> Specifically, we adopt the following protocol: we first pretrain on the large-scale molecular graph dataset **ChEMBL**, then fine-tune on the downstream **HIV** dataset, and report the test performance. Since HIV is a representative graph-level molecular property prediction task, this experiment directly evaluates the **transferability** of R-GFM in the molecular graph setting.
>
> The additional results are as follows:
>
> |Model|Test AUC|
> |-|-|
> |MDGFM|0.7563|
> |SAMGPT|0.7664|
> |RiemannGFM|0.7804|
> |GCOPE|0.7230|
> |**R-GFM**|**0.7849**|
>
> As shown above, R-GFM achieves the **best** Test AUC of **0.7849**, outperforming the strongest baseline RiemannGFM (**0.7804**) as well as MDGFM, SAMGPT, and GCOPE. This result suggests that the core design of R-GFM—multi-scale subgraph/GoG modeling and dynamic Riemannian routing—is not limited to node-level tasks, but also provides **consistent benefits in the graph-level setting**.
>
> In the revised manuscript, we will include these results in the experimental section to demonstrate the effectiveness of our method on graph-level tasks.
>
> # Response to W3&Q3 (Comparison with Mixed-hop Aggregation Methods)
>
> Thank you for valuable advice. Mixed-hops aggregation method such as MixHop [1] merely concatenate information from different neighborhood hops during message passing. As a result, they **fail to capture the intrinsic topological dependencies** across different hop scales, nor can they adaptively adjust the receptive field to task-specific structural requirements.
>
> In contrast, R-GFM constructs adaptive-hop subgraphs, organizes them into a Graph-of-Graphs (GoG) to explicitly model cross-scale relations to better capture and reason over heterogeneous structural contexts across scales.
>
> Following your suggestion, we additionally included **MixHop** as a new baseline, under the same 1-shot node classification setting as our main experiments. As shown in the Table below, **R-GFM still achieves the best performance on all 8 test datasets**.
>
> |Model|Wisconsin|Cornell|Citeseer|Cora|Pubmed|Computers|Photos|Texas|
> |-|-|-|-|-|-|-|-|-|
> |MixHop|29.82±9.32|17.14±5.54|21.35±3.67|21.46±5.95|43.57±4.05|29.77±15.57|32.77±9.26|26.09±15.93|
> |R-GFM|**35.41±7.29**|**36.71±9.92**|**57.54±9.49**|**49.50±3.97**|**49.80±5.38**|**52.30±3.33**|**61.08±5.26**|**32.36±12.10**|
>
> We will incorporate this discussion into the related work section and present the corresponding experimental results in the experiments section of the revised manuscript to provide a more comprehensive evaluation against existing multi-hop paradigms.
>
> ---
>
> **References:**
>
> [1] MixHop: Higher-Order Graph Convolutional Architectures via Sparsified Neighborhood Mixing. ICML 19

---

> > ### Author Rebuttal · Reviewer_uePc · 2026-04-02
> >
> > I thank the authors for the detailed rebuttal addressing my questions. I appreciate the additional experiments. The results for molecular property prediction let me wonder though: Compared to the reported results on the public open graph benchmark platform (ogb : https://ogb.stanford.edu/docs/leader_graphprop/#ogbg-molhiv) an ROCAUC of 0.78 is not competitive to simple, non-pretrained GNNs. Can the author please comment on that?

---

> > > ### Author Response · Authors · 2026-04-04
> > >
> > > We sincerely thank you for acknowledging our rebuttal. We appreciate your keen insight regarding the OGB leaderboard.
> > >
> > > To alleviate your concerns, we provide clarifications from the following two perspectives:
> > >
> > > **(1) The problem settings for pretrained models and simple, non-pretrained GNNs are fundamentally different.**
> > >
> > > While the current pretrained models may yield lower performance compared to simple, non-pretrained GNNs, it is important to note that their problem settings are **fundamentally different**.
> > >
> > > Non-pretrained GNNs on the OGB leaderboard are often heavily optimized and trained from scratch for a single specific task with abundant labeled data, but pre-trained models aim to **generalize across diverse downstream tasks and domains**. Specifically, the advantages of pre-training over non-pre-training lie in the following aspects:
> > >
> > > - **Universal Representation and Transferability:** Non-pretrained GNNs are usually task-specific and prone to overfitting the training distribution of a **single dataset**. In contrast, pre-trained models capture intrinsic structural from **massive unlabeled data**, allowing them to extract universal representations that can be **transferred** across various downstream tasks and domains.
> > >
> > > - **Data Efficiency (Few-shot/Zero-shot Learning):** Non-pretrained GNNs heavily rely on **labeled data** to achieve high performance. However, in real-world molecular property prediction, labeled data is often **scarce or expensive** to obtain.
> > >
> > > **(2) When existing simple, non-pretrained GNNs are adapted to the pre-training setting, they are outperformed by R-GFM**.
> > >
> > > We further evaluate GFM baselines and the current top five state-of-the-art methods from the OGB leaderboard (including **Multi-RF Fusion + Multi-GNN**,**HyperFusion**, **PAS+FPs**, **HIG**, and **DeepAUC** ) in 1-shot, 3-shot, and 5 shot settings.
> > >
> > > Specifically, we first pretrain on the large-scale molecular graph dataset **ChEMBL**, and then adapt to the downstream HIV task using only **1 or 3 or 5 labeled example per class**. This setting is aligned with the pre-training setting, i.e., whether the learned graph foundation model can transfer across domains and remain effective under extremely limited supervision.
> > >
> > > As shown in the tables below, R-GFM consistently outperforms all GFM baselines as well as the top five state-of-the-art methods on the OGB leaderboard.
> > >
> > > **Remark:**  In our initial response, we reported the results of **full-shot fine-tuning** for GFMs (i.e., fine-tuning on the entire downstream dataset). We have now additionally provided the results under 1-shot, 3-shot, and 5-shot settings.
> > >
> > > ### 1-shot
> > > |Method|Test AUC|
> > > |-|-|
> > > |Multi-RF Fusion + Multi-GNN|0.5498|
> > > |HyperFusion|0.5101|
> > > |PAS+FPs|0.5003|
> > > |HIG|0.5433|
> > > |DeepAUC|0.5172|
> > > |MDGFM|0.6101|
> > > |SAMGPT|0.6170|
> > > |RiemannGFM|0.6175|
> > > |GCOPE|0.5825|
> > > |**R-GFM**|**0.6473**|
> > >
> > > ### 3-shot
> > > |Method|Test AUC|
> > > |-|-|
> > > |Multi-RF Fusion + Multi-GNN|0.5764|
> > > |HyperFusion|0.5285|
> > > |PAS+FPs|0.5948|
> > > |HIG|0.5913|
> > > |DeepAUC|0.5910|
> > > |MDGFM|0.6169|
> > > |SAMGPT|0.6248|
> > > |RiemannGFM|0.6273|
> > > |GCOPE|0.6354|
> > > |**R-GFM**|**0.6557**|
> > >
> > > ### 5-shot
> > > |Method|Test AUC|
> > > |-|-|
> > > |Multi-RF Fusion + Multi-GNN|0.5913|
> > > |HyperFusion|0.5909|
> > > |PAS+FPs|0.6082|
> > > |HIG|0.5973|
> > > |DeepAUC|0.6068|
> > > |MDGFM|0.6365|
> > > |SAMGPT|0.6424|
> > > |RiemannGFM|0.6368|
> > > |GCOPE|0.6536|
> > > |**R-GFM**|**0.6648**|
> > >
> > > We agree that improving the cross-domain pre-training performance on **graph classification** tasks is highly meaningful (similar to how our model already outperforms simple, non-pretrained GNNs on **node classification tasks**, as demonstrated in **Table 1 of the manuscript**). Given the consistent limitations of existing graph foundation models in **graph classification**, we consider this an important direction for future work.

---

### Official Review · Reviewer_Ymod · 2026-03-09

**Soundness:** 2
**Presentation:** 2
**Significance:** 3
**Originality:** 3
**Overall Recommendation:** 4
**Confidence:** 4

**Summary:**

This paper aims to tackle cross-domain graph foundation models and argues that fixed-hop subgraph pretraining creates a structural scale mismatch across tasks.  The authors propose R-GFM which builds a multi-hop Graph-of-Graphs and routes each GoG through dynamic Riemannian experts (hyperbolic / Euclidean / spherical). The author also claims tighter theory bounds and reports broad experiments (few-shot node classification, link pred, robustness, scaling).

**Compliance With Llm Reviewing Policy:**

Affirmed.

**Final Justification:**

Thanks for the author's follow-up response and new clarification. After the new rounds of response, I rechecked the paper, and now my main concerns are addressed, so I'd like to raise my ratings. Please include these additional theoretical patches and experimental results in your future revision.

**Key Questions For Authors:**

1. Please provide the exact missing statement for Assumption ?? and clarify whether it holds in your experimental setup.

2. For the “strictly tighter” bound, can you clearly list the exact conditions under which strict improvement is guaranteed?

3. Could you add a short section that checks key theoretical quantities empirically, even approximately?

**Limitations:**

No. The current draft does not clearly state where the theory may fail or become less informative.

**Strengths And Weaknesses:**

**Strengths**
- The problem is real and important. Fixed-hop assumptions do hurt transfer in practice.
- Personally, I think the method design is coherent, multi-scale GoG + geometry-aware routing is not just random stacking.
- Empirical coverage is fairly broad, with robustness and ablations.

**Weaknesses**
- When I read the paper, I found that there's a pretty serious factual issue in the theory appendix, it contains unresolved placeholders (e.g. Assumption ?? in Line 1501). That makes the “strictly tighter bound” claim hard to fully trust as written.
- I quickly go through the paper and I think some claims feel stronger than what is clearly justified in the main text, especially around when the theory guarantees actually apply.

---

> ### Author Rebuttal · Authors · 2026-03-30
>
> # Response to W1&Q1 (Missing Assumption)
>
> Sorry for the confusion. We inadvertently omitted an assumption necessary for comparing R-GFM encoder and MDGFM encoder. The omitted assumption is provided below.
>
> **Assumption 1 (Reference-space comparison condition)**
> Relative to the same model-independent reference space, R-GFM yields a representation geometry that brings the target domain closer to the source-domain family than MDGFM does.
>
> Under this comparison condition, the bound quantity in Theorem 3.5 is smaller for R-GFM than for MDGFM. We will add this condition explicitly in Appendix J, Step 3.
>
> **Justification.** The practice of making assumptions about intermediate relationships, as in Assumption 1, is widely used in theoretical analyses for cross-domain transfer [1,2,3].
>
> **Empirically**, we verify Assumption 1 using $\hat{\gamma}$, the distance from the target-domain centroid to the convex hull of the source-domain centroids in the learned latent space. R-GFM yields a smaller $\hat{\gamma}$ than MDGFM as shown in the table below, indicating that R-GFM brings the target domain closer to the source-domain family. This provides empirical support for Assumption 1.
>
> |Datasets|Model|$\hat{\gamma}$|
> |-|-|-|
> |wisconsin|R-GFM|**0.0720**|
> ||MDGFM|0.1573|
> |cornell|R-GFM|**0.0789**|
> ||MDGFM|0.1577|
> |citeseer|R-GFM|**0.0321**|
> ||MDGFM|0.2009|
> |cora|R-GFM|**0.0352**|
> ||MDGFM|0.3979|
> |pubmed|R-GFM|**0.3760**|
> ||MDGFM|1.0006|
> |computers|R-GFM|**0.0709**|
> ||MDGFM|0.2605|
> |photo|R-GFM|**0.0752**|
> ||MDGFM|0.2965|
> |texas|R-GFM|**0.0786**|
> ||MDGFM|0.2235|
>
> [1] A Discriminative Technique for Multiple-Source Adaptation. ICML 21
>
> [2] A Theory of Multiple-Source Adaptation with Limited Target Labeled Data. AISTATS 21
>
> [3] MADG: Margin-based Adversarial Learning for Domain Generalization. NeurIPS 23
>
> ---
>
> # Response to Q2&W2 (Exact Conditions)
>
> Thank you for your advice. For the “strictly tighter” bound in Theorem 3.5, the conditions are concluded as follows:
>
> **Condition 1**: The discrepancy measure $d(\cdot,\cdot)$ satisfies: 1) the triangle inequality, and 2) convexity with respect to mixing, as shown in Appendix J, Eq. (48).
>
> **Condition 2**: There exists a reference encoder $\phi^\star$, fixed independently of the compared models, such that in this reference space the target-domain embedding distribution can be approximated by a convex combination of the source-domain embedding distributions. This is provided in Appendix J, Eq. (49).
>
> Based on these conditions, we can derive that the domain generalization error upper bound of R-GFM is strictly tighter than that of MDGFM.
>
> We will explicitly state the exact conditions in the revised manuscript.
>
> ---
>
> # Response to Q3&W2 (Empirical Justification)
>
> Thanks for your valuable advice. Following your suggestion, we computed an observable proxy of the core quantity in **Theorem 3.5**. For a fair comparison, we align the input node features for all evaluated models by using the same leave-one-dataset-out protocol and the same normalized latent space. Consequently, the empirical comparison reduces to evaluating the model-dependent term, which can be computed as $\hat{B} = \hat{\gamma} + \frac{1}{2}\hat{\rho}.$
>
> Under this unified setting, the relative comparison between R-GFM and MDGFM is reduced to comparing $\hat{B}$. A smaller $\hat{B}$ indicates a smaller target-to-source mismatch and a tighter source-domain geometry [4], and is therefore consistent with a tighter bound in **Theorem 3.5**. The detailed definitions of these quantities are as follows:
>
> - (1) the distance from the target-domain centroid to the convex hull of the source-domain centroids, denoted by $\hat{\gamma}$;
> - (2) the diameter of the set of source-domain centroids, denoted by $\hat{\rho}$;
> - (3) the combined quantity $\hat{B}=\hat{\gamma}+\frac{1}{2}\hat{\rho}$.
>
> The experimental results (as shown in the table below) show that R-GFM achieves a smaller combined quantity $\hat{B}$ on all 8 datasets, indicating consistency with the conclusions of **Theorem 3.5**.
>
> We will add the empirical results to the revised manuscript to better bridge our theoretical analysis with empirical observations.
>
> |Datasets|Model|$\hat{\gamma}$|$\hat{\rho}$|$\hat{B}=\hat{\gamma}+\hat{\rho}/2$|
> |-|-|-|-|-|
> |wisconsin|R-GFM|**0.0720**|0.5073|**0.3257**|
> ||MDGFM|0.1573|**0.3900**|0.3523|
> |cornell|R-GFM|**0.0789**|0.5247|**0.3413**|
> ||MDGFM|0.1577|**0.3876**|0.3515|
> |citeseer|R-GFM|**0.0321**|0.5054|**0.2848**|
> ||MDGFM|0.2009|**0.3269**|0.3643|
> |cora|R-GFM|**0.0352**|**0.5117**|**0.2910**|
> ||MDGFM|0.3979|0.5971|0.6964|
> |pubmed|R-GFM|**0.3760**|**0.1764**|**0.4642**|
> ||MDGFM|1.0006|0.4488|1.2250|
> |computers|R-GFM|**0.0709**|0.5039|**0.3228**|
> ||MDGFM|0.2605|**0.4694**|0.4952|
> |photo|R-GFM|**0.0752**|0.5042|**0.3274**|
> ||MDGFM|0.2965|**0.4120**|0.5025|
> |texas|R-GFM|**0.0786**|0.5336|**0.3454**|
> ||MDGFM|0.2235|**0.4334**|0.4402 |
>
> [4] Multi-Domain Graph Foundation Models: Robust Knowledge Transfer via Topology Alignment. ICML 25.

---

> > ### Author Rebuttal · Reviewer_Ymod · 2026-04-03
> >
> > Thank you for the rebuttal. I appreciate that the authors clarified the missing assumption and added empirical proxies for the bound-related quantities. That said, my main concern is only partially resolved. In particular, the newly introduced “reference-space comparison condition” appears quite strong and, at least in its current form, is close to assuming the desired ordering between R-GFM and MDGFM rather than deriving it from milder conditions. The additional empirical quantities are helpful as supporting evidence, but they do not fully close this gap at the theorem level. Since this concern is tied to a central theoretical claim of the paper, I do not think it can be fully resolved within a short rebuttal, so I am maintaining my original score.

---

> > > ### Author Response · Authors · 2026-04-06
> > >
> > > Thank you for acknowledging that we have addressed some of your concerns. To address your remaining concerns, we provide more theoretical analysis as follows:
> > >
> > > Following your suggestions, we provide an additional theoretical comparison between R-GFM and MDGFM under a **weaker assumption** than before, as detailed below:
> > >
> > > ---
> > > ## Theorem 1 (Tighter Target-Domain Error Bound of R-GFM)
> > > Let $\Phi\_M$ denote MDGFM, and let $\Phi\_R$ denote R-GFM.
> > >
> > > For any encoder $\phi$, define the comparison functional
> > > $$
> > > \mathcal{B}(\phi):=\underbrace{\Delta\_{\mathrm{fit}}(\phi)}\_{\text{fitting term}}+\underbrace{\Delta\_{\mathrm{scale}}(\phi)}\_{\text{scale mismatch term}}+\underbrace{\Delta\_{\mathrm{dom}}(\phi)}\_{\text{cross-domain discrepancy term}}.
> > > $$
> > >
> > > Assume that the assumption below holds, and define
> > > $$
> > > \eta := \varepsilon\_g-\varepsilon\_d > 0,
> > > $$
> > > where $\varepsilon\_g>0$ is the source-side improvement margin (including the fitting term, the scale-mismatch term) and $\varepsilon\_d\ge 0$ is the residual target-side discrepancy gap (cross-domain discrepancy term).
> > >
> > > Let
> > >
> > > $$
> > > \mathcal{B}\_M^\star := \inf_{\phi\in\Phi\_M}\mathcal{B}(\phi),
> > > \qquad
> > > \mathcal{B}\_R^\star := \inf_{\phi\in\Phi\_R}\mathcal{B}(\phi).
> > > $$
> > >
> > > Then
> > >
> > > $$
> > > \mathcal{B}\_R^\star
> > > \le
> > > \mathcal{B}\_M^\star - \eta
> > > <
> > > \mathcal{B}\_M^\star.
> > > $$
> > >
> > > Therefore, the target-domain error upper bound over
> > > $\Phi\_R$ is strictly tighter than the target-domain error over $\Phi\_M$.
> > >
> > > ---
> > > ## Assumption (Negligible Relative Domain Bias)
> > >
> > > Based on the domain adaptation and domain generalization theory [1][2], we show that for any encoder $\phi$, the target-domain error can be upper bounded by a source-side fitting term, a statistical complexity term, a graph-specific scale-mismatch term, and a cross-domain discrepancy term:
> > >
> > > $$
> > > \epsilon
> > > \le
> > > \Delta\_{\mathrm{fit}}(\phi)
> > > +
> > > \Delta\_{\mathrm{scale}}(\phi)
> > > +
> > > \Delta\_{\mathrm{dom}}(\phi).
> > > $$
> > >
> > > Equivalently, with
> > >
> > > $$
> > > \mathcal{B}(\phi):=
> > > \Delta\_{\mathrm{fit}}(\phi)
> > > +
> > > \Delta\_{\mathrm{scale}}(\phi)
> > > +
> > > \Delta\_{\mathrm{dom}}(\phi),
> > > $$
> > >
> > > we have
> > >
> > > $$
> > > \epsilon\le \mathcal{B}(\phi).
> > > $$
> > >
> > > Following previous works [3][4], we assume that the domain bias term $\Delta\_{\mathrm{dom}}(\phi)$ can be ignored for both $\phi \in \Phi\_M$ and $\phi \in \Phi\_R$. As a result, $\Delta\_{\mathrm{dom}}(\phi)$ remains small for $\phi\in\Phi\_M$ as well as for $\phi\in\Phi\_R$.
> > >
> > > In other words, we assume that the discrepancy gap between the two hypothesis classes is negligible:
> > > $$
> > > \left|\Delta\_{\mathrm{dom}}(\phi\_M)-\Delta\_{\mathrm{dom}}(\phi\_R)\right|\approx 0,
> > > \qquad
> > > \phi\_M\in\Phi_M,\ \phi\_R\in\Phi\_R.
> > > $$
> > >
> > > ---
> > >
> > > **Detailed proof is provided in [link](https://anonymous.4open.science/r/proof-327B/proof.pdf).**
> > >
> > > ---
> > >
> > > ## Assumption Justification
> > >
> > > **This assumption is weaker than before** because it relaxes **the reference-space comparison condition with strict target-source alignment** (as shown in our initial response) into a **negligible relative domain bias assumption** [1-4], which is widely used as **a fundamental and mild condition** in existing works [1-4].
> > >
> > > **Empirically**, we validate these assumptions as follows. The results support our claim that, $\Delta\_{\mathrm{dom}}(\phi)$, remains negligibly small across all datasets, indicating that the cross-domain discrepancy is not the main factor affecting the overall bound, **which supports the assumption**. In contrast, R-GFM consistently achieves smaller $\mathcal{B}(\phi)$ than MDGFM, **which directly supports Theorem 1**.
> > >
> > > |Target|Model| $\Delta\_{\mathrm{fit}}(\phi)$ |$\Delta\_{\mathrm{scale}}(\phi)$|$\Delta\_{\mathrm{dom}}(\phi)$|$\mathcal{B}(\phi)$|
> > > |-|-|-|-|-|-|
> > > |Cora|R-GFM|**0.335**|**0.210**|**0.003**|**0.548**|
> > > ||MDGFM|0.413|0.237|0.004|0.654|
> > > |Chameleon|R-GFM|**0.313**|**0.146**|0.003|**0.462**|
> > > ||MDGFM|0.445|0.371|**0.001**|0.817|
> > > |Texas|R-GFM|**0.325**|0.164|**0.005**|**0.494**|
> > > || MDGFM|0.493|**0.187**|0.007|0.687|
> > > |Photo|R-GFM|**0.354**|**0.203**|0.004|**0.561**|
> > > ||MDGFM|0.509|0.360|**0.003**|0.872|
> > > ## Theoretical Improvements in our work
> > >
> > > Here, we conclude our **theoretical improvements** compared to previous GFM works. Prior GFMs either do not provide a theoretical analysis (e.g., **GCOPE** and **SAMGPT**), or provide theory only for their own mechanism without directly comparing one GFM against another at the theorem level (e.g., **RiemannGFM** and **MDGFM**).
> > >
> > > In contrast, our work introduces a **new perspective** for conducting comparative theoretical analyses among different GFMs based on the **target-domain error** and prove the **theoretical superiority** of our proposed method over MDGFM.
> > >
> > > We will incorporate these into the revised manuscript to strengthen the theoretical contributions of our paper.
> > >
> > > ---
> > > **References**
> > >
> > > [1] Domain Generalization via Invariant Feature Representation. ICML 13
> > >
> > > [2] Bridging Theory and Algorithm for Domain Adaptation. ICML 19
> > >
> > > [3] Measuring the Robustness of NLP Models to Domain Shifts. EMNLP 24
> > >
> > > [4] In Search of Lost Domain Generalization. ICLR 21

---

### Official Review · Reviewer_tG6C · 2026-03-14

**Soundness:** 3
**Presentation:** 3
**Significance:** 3
**Originality:** 3
**Overall Recommendation:** 4
**Confidence:** 4

**Summary:**

This work proposes R-GFM, a Riemannian Graph-of-Graphs-based foundation model that treats structural scale as a first-class citizen in modeling, addressing the challenge of building foundation models that generalize across diverse graph structures and domains.
Both theoretical analysis and empirical experiments are conducted to demonstrate the effectiveness of the proposed method.

**Compliance With Llm Reviewing Policy:**

Affirmed.

**Final Justification:**

My original assessment raised three concerns regarding the miss baseline, theoretical analysis, and the scalability of the proposed method.
The authors' rebuttal has fully addressed my concern, and thus I raise my score.

**Key Questions For Authors:**

See above.

**Limitations:**

YEs

**Strengths And Weaknesses:**

Strengths
1. The overall manuscript is well organized, and the presentation is good.
2. The source code is provided to ensure reproduciblitiy.
3. Theorems 3.2 and 3.3 provide new insights into multi-hop fusion and GoG edge construction that don't appear in prior work.

Weakness

1. Missing some key baseline methods that irrelevant to MPNNs, such as GPM [1]
2. Theorem 3.3 seems based on strong assumptions. The gap between theoretical conditions and practical applicability is concerning.
3. What is the scalability of the proposed method? How does it perform on large size graphs, such as graphs that contain billion-level nodes?


[1] Generative graph pattern machine. ICML'25

---

> ### Author Rebuttal · Authors · 2026-03-31
>
> # Response to W1
> Thank you for your advice. GPM [1] and its self-supervised extension G2PM [2] learn from substructure sequences instead of MPNNs. However, they do not explicitly model cross-scale subgraph relations. In contrast, R-GFM organizes multi-hop subgraphs into a GoG equipped with a Riemannian MoE to better capture task-dependent receptive fields.
>
> Following your advice, we added them as baselines and report their 1-shot node classification results. R-GFM achieves the best model quality as shown below.
>
> We will include the results in the revised manuscript.
>
> |Model|Wisconsin|Cornell|Citeseer|Cora|Pubmed|Computers|Photos|Texas|
> |-|-|-|-|-|-|-|-|-|
> |GPM|29.2|27.8|29.9|39|35.8|48.6|37.9|27.7|
> |G2PM|34.7|33|31|40|49.7|51.9|59.9|30.4|
> |R-GFM|35.4|36.7|57.5|49.5|49.8|52.3|61.1|32.4|
>
> [1] Beyond Message Passing: Neural Graph Pattern Machine. ICML 25
>
> [2] Generative Graph Pattern Machine. NeurIPS 25
>
> ---
> # Response to W2
> Thanks for your comments. We would like to clarify that the assumptions in Theorem 3.3 are **not overly restrictive**, as they are **commonly adopted** in the existing literature.
>
> **Assumption H.1** adopts a standard second-order noise model [3] to assume a weak correlation among different-hop subgraph embeddings, as commonly used in [3,4,5]. To relate it to practice, we empirically evaluate these correlations using the ratio of cross-hop covariance to within-hop variance based on [6]. Since a smaller ratio indicates a weaker inter-hop correlation, the consistently low ratios observed below empirically justify our assumption.
>
> |Dataset|Avg. cross-hop cov./within-hop var.|
> |-|-|
> |Photo|0.001|
> |Cora|0.002|
> |Chameleon|0.0005|
> |Texas|0.0006|
>
> **Assumption H.2** indicates that more similar subgraphs should incur smaller mismatch when aggregated, which is consistent with the smoothness principle [7]. Practically, we utilize Spearman's ρ(pred) and (proto) to quantify this mismatch. Concretely, we compute the Spearman correlation between within-node hop-pair similarity and either prediction mismatch or class-prototype mismatch.
>
> As shown below, the negative values on 4 datasets indicate that higher subgraph similarity is associated with lower mismatch.
>
> ||ρ(pred)|ρ(proto)|
> |-|-|-|
> |Photo|-0.6|-0.3|
> |Cora|-0.6|-0.4|
> |Texas|-0.5|-0.2|
> |Chameleon|-0.2|-0.1|
>
> **Assumption H.3** establishes that similarity-based selective connectivity is superior to having either no cross-hop interaction or full connectivity. This assumption is commonly used in the analysis of over-smoothing  [8].
>
> Empirically, the results as below demonstrate that similarity-based selective connectivity (i.e., ratio = 0.6) achieves higher accuracy compared to both overly sparse (i.e., ratio = 0.2) and fully dense (i.e., ratio = 1.0) connections.
>
> |Dataset|ratio|Acc.%|
> |-|-|-|
> |Cora|0.2|45|
> ||0.4|38|
> ||0.6|**49**|
> ||0.8|41|
> ||1.0|44|
> |Photo|0.2|49|
> ||0.4|51|
> ||0.6|**61**|
> ||0.8|60|
> ||1.0|44|
> |Chameleon|0.2|22|
> ||0.4|23|
> ||0.6|**23**|
> ||0.8|22|
> ||1.0|21|
> |Texas|0.2|23|
> ||0.4|26|
> ||0.6|**32**|
> ||0.8|31|
> ||1.0|23|
>
> [3] A Unified View on Graph Neural Networks as Graph Signal Denoising. CIKM 21
>
> [4] Bias-Variance Tradeoff of Graph Laplacian Regularizer. IEEE Signal Process. Lett. 17
>
> [5] Rethinking Semi-Supervised Imbalanced Node Classification from Bias-Variance Decomposition. NeurIPS 23
>
> [6] Testing for High-Dimensional White Noise Using Maximum Cross-Correlations. Biometrika 17.
>
> [7] Non-Local Means Denoising. IPOL 11
>
> [8] Not too little, not too much: a theoretical analysis of graph (over)smoothing. NeurIPS 22
>
> ---
> # Response to W3
> Following your advice, we report training time vs. data scale (i.e., node number), as shown below. We can observe that R-GFM demonstrates good scalability, meaning the training time scales linearly as the data size grows.
>
> Compared with two full-graph training GFM baselines, SAMGPT and MDGFM, R-GFM has a significant advantage in scalability. SAMGPT and MDGFM encounter OOM issues when the node number reaches 169K, whereas R-GFM successfully completes the training because of its sampling mechanism.
>
> Compared with the other two sampling-based baselines, ours delivers the best quality while maintaining comparable linear scalability.
>
> Regarding the billion-scale graphs, we would like to clarify that the largest dataset fully utilizing all training nodes in existing open-source GFM works is ogbn-products (~2M nodes), which is included in our analysis below. For billion-scale graphs, extrapolating from our scaling curve, the training time is approximately 15 days on 8 A100s, which is in the same order of magnitude as the fastest baseline (17 days for RiemannGFM on 8 A100s).
>
> We will incorporate these results in the revised manuscript.
>
> |Node Num/Time(s)|R-GFM|SAMGPT|MDGFM|GCOPE|RiemannGFM|
> |-|-|-|-|-|-|
> |617|22|79|20|58|47|
> |3K|92|106|27|183|100|
> |6K|89|134|31|364|166|
> |26K|595|185|103|1508|536|
> |40K|623|243|142|2508|1421|
> |47K|1187|282|155|3162|2740|
> |169K|3619|OOM|OOM|9111|4296|
> |2M|50294|OOM|OOM|105508|56866|

---

> > ### Author Rebuttal · Reviewer_tG6C · 2026-04-02
> >
> > Thank the author for the detailed rebuttal. My concerns have been fully addressed. Therefore, I raise my score.

---

> > > ### Author Response · Authors · 2026-04-04
> > >
> > > We sincerely appreciate your confirmation that our rebuttal has successfully addressed all your concerns, and we are truly grateful for your updated positive score.
> > >
> > > Thank you for your time and effort in reviewing our paper.

---

### Official Review · Reviewer_SyZJ · 2026-03-16

**Soundness:** 3
**Presentation:** 3
**Significance:** 3
**Originality:** 3
**Overall Recommendation:** 4
**Confidence:** 4

**Summary:**

This paper proposes R-GFM (Riemannian Graph-of-Graphs), a graph foundation model that addresses the scale-mismatch problem of fixed-hop pretraining. Specifically, the proposed method constructs an adaptive-hop graph-of-graphs and encodes it with a dynamic mixture of Riemannian experts. The method has a clear motivation, and is theoretically supported. Experimental results show that the proposed method shows strong results across cross-domain node classification and link prediction benchmarks.

**Compliance With Llm Reviewing Policy:**

Affirmed.

**Final Justification:**

The authors‘ responses have addressed my comments. I maintain my slightly positive score. I believe this paper makes a valuable contribution towards geometric graph foundation model.

**Key Questions For Authors:**

See weaknesses/questions mentioned above

**Limitations:**

Yes

**Strengths And Weaknesses:**

Strengths:
S1. The paper tackles an important problem in graph foundation models: the mismatch between fixed-hop pretraining and task-dependent structural scales.
S2. The proposed Graph-of-Graphs formulation is interesting and gives the method a clear multi-scale perspective.
S3. The dynamic Riemannian expert design makes the framework more expressive than standard Euclidean encoders.

Weaknesses/Questions:
W1. Though I acknowledge the overall framework of the proposed method is relatively clearly designed, some components (e.g., the expert selection and routing design) appear somewhat heuristic. The authors could further elaborate on these designs.
W2. The authors provide some theoretical results, e.g., theoretically outperforms MDGFM, which is commendable, but I wonder whether such analyses could be extended to RiemannGFM, which is a more related work, in my opinion.
W3. In the experiments, the authors compare a range of GFM baselines on a range of benchmark datasets, but more recent baselines and more large-scale benchmarks could further enhance the paper.
W4. More relevant literature on Graph-of-Graph should be surveyed in the related works.

---

> ### Author Rebuttal · Authors · 2026-03-30
>
> # Response to W1
> Thanks for your advice. We would like to clarify that the expert selection and routing design are grounded in sound theoretical motivations rather than being purely heuristic.
>
> For the **expert selection**, existing MoE-based methods (e.g., GraphMoRE [1]) employ a manually specified number of experts, which requires extensive trial-and-error tuning and fails to adapt to various graph datasets. In contrast, our dynamical candidate Expert can adaptively determine the optimal number of experts and achieve a tighter excess risk upper bound, as proved in **Theorem 3.4**.
>
> For the **routing design,** existing works  (e.g., RiemannGFM)  typically employ hand-crafted, heuristic static routing strategies. In contrast, our routing design is dynamic and learnable, which aligns with the dynamic GoG construction process. Based on the GoG routing design, R-GFM achieves a lower Domain Generalization Error Bound compared to MDGFM, as proved in **Theorem 3.5**.
>
> **Experimentally**, the ablation experiments in **Fig. 7 and 8** of the manuscript demonstrate the effectiveness of these two techniques.
>
> [1] Graphmore: Mitigating topological heterogeneity via mixture of riemannian experts. AAAI 2025
>
> ---
>
> # Response to W2
> Thanks for your advice. We would like to clarify that our theoretical analysis can be extended to RiemannGFM. RiemannGFM can be viewed as a degenerate special case of our framework, which uses a fixed Riemannian manifold choice. In **Theorem 3.4**, we established the excess risk upper bounds for both the degenerate special case and our proposed method. Base on these, we establish the following theorem.
>
> **Theorem 1.** R-GFM achieves an excess risk upper bound that is no worse than RiemannGFM.
>
> **Proof:** RiemannGFM is a degenerate special case of our framework, where the candidate expert set is restricted to a single fixed Riemannian manifold and dynamic routing is disabled. Therefore, it corresponds to a manually fixed expert configuration. By Theorem 3.4, the excess risk upper bound of our dynamic expert selection strategy is no worse than that of any manually fixed expert configuration. Hence, this theorem is proven.
>
> ---
>
> # Response to W3
> Thanks for your advice. GPM [2] and G2PM [3] learn from substructure sequences and do not explicitly model cross-scale subgraph relations. In contrast, R-GFM organizes multi-hop subgraphs into a GoG equipped with a Riemannian MoE to better capture task-dependent receptive fields.
>
> We included their 1-shot node classification results below. R-GFM achieves the best performance.
>
> ||Wisconsin|Cornell|Citeseer|Cora|Pubmed|Computers|Photos|Texas|
> |-|-|-|-|-|-|-|-|-|
> |GPM|29.2|27.8|29.9|39|35.8|48.6|37.9|27.7|
> |G2PM|34.7|33|31|40|49.7|51.9|59.9|30.4|
> |R-GFM|35.4|36.7|57.5|49.5|49.8|52.3|61.1|32.4|
>
> In terms of scalability, the key difference is the training paradigm. In particular, R-GFM adopts a sampling-based training pipeline, which leads to near-linear scaling of training time with respect to the number of nodes, as shown in the table below.
>
> Full-graph GFM baselines (e.g., SAMGPT and MDGFM) run into OOM issues once the graph size reaches 169K nodes, whereas R-GFM remains trainable under the same setting due to its sampling mechanism.
>
> Compared with other sampling-based baselines, R-GFM maintains comparable scalability while achieving the best model quality, indicating a better trade-off between efficiency and quality.
>
> We will include these experiments in the revised manuscript.
>
> |Node Num./Time(s)|RGFM|SAMGPT|MDGFM|GCOPE|RiemannGFM|
> |-|-|-|-|-|-|
> |617|22|79|20|58|47|
> |3K|92|106|27|183|100|
> |6K|89|134|31|364|166|
> |26K|595|185|103|1508|536|
> |40K|623|243|142|2508|1421|
> |47K|1187|282|155|3162|2740|
> |169K|3619|OOM|OOM|9111|4296|
> |2M|50294|OOM|OOM|105508|56866|
>
> [2] Beyond Message Passing: Neural Graph Pattern Machine. ICML 25
>
> [3] Generative Graph Pattern Machine. NeurIPS 25
>
> ---
>
> # Response to W4
> Thanks for your advice.
>
> Existing GoG methods are not directly applicable to our setting. Existing GoG studies are developed for static scenarios, where the GoG is constructed over a fixed collection of graph instances, as in G2GNN [4] and ImbGNN [5]. In particular, G2GNN constructs the GoG based on kernel similarity between graphs. ImbGNN employs a similarity-aware random walk to extract subgraphs, which incurs a high computational cost during this extraction step.
>
> In contrast, our framework is dynamic and efficient. We construct the GoG with a memory-aware strategy that adaptively expands the hop range. Moreover, unlike methods that require expensive similarity computation online and thus incur repeated costs during training or inference, our GoG is built from sampled multi-hop subgraphs with similarity computed once offline, making the construction more efficient.
>
> [4] Imbalanced Graph Classification via Graph-of-Graph Neural Networks. CIKM 22
>
> [5] When imbalance meets imbalance: Structure-driven learning for imbalanced graph classification. WWW 24

---

> > ### Author Rebuttal · Reviewer_SyZJ · 2026-04-02
> >
> > The authors‘responses’ have addressed my comments. I maintain my positive score.

---

> > > ### Author Response · Authors · 2026-04-04
> > >
> > > Thank you for confirming that our responses have addressed all your concerns.  We truly appreciate your maintained positive score. Given that the concerns have been successfully resolved, we would be highly grateful if you could kindly consider bumping up your score.
> > >
> > > Thank you for your time and effort in reviewing our paper.

---

### Decision · Program_Chairs · 2026-04-30

**Decision:**

Accept (regular)

**Comment:**

In the context of Graph foundation models (GFMs), the paper introduces R-GFM, a model designed to handle varying structural scales using a multi-scale graph-of-graphs approach combined with Riemannian representations. It addresses a key limitation in existing methods, where fixed-hop assumptions reduce performance when transferring across tasks with different structural requirements. The proposed framework is both novel and coherent, integrating multi-scale representations with geometry-aware processing through dynamic Riemannian experts, which makes it more expressive than standard Euclidean models.
The work is well presented and clearly organized, and it provides source code to support reproducibility. It also contributes new theoretical insights into multi-hop fusion and graph-of-graphs construction. The problem it tackles is practical and important, as rigid subgraph sampling strategies often limit the effectiveness of graph foundation models. Empirical evaluation is good, covering multiple datasets and domains, and shows consistent improvements over existing baselines. Additional ablation and robustness studies further demonstrate the value of the core design choices, supporting the effectiveness and originality of the approach. Removal of assumptions for comparison between R-GFM and MDGFM would be desirable.
Rebuttal contributed to clarify the main issues raised by reviewers, and also to improve the quality of the paper.